

# Comprehensive mass spectrometric analysis of unprecedented high levels of carbonaceous aerosol particles long-range transported from wildfires in the Siberian Arctic

Eric Schneider[1,2], Hendryk Czech[1], Olga Popovicheva[3], Marina Chichaeva[4], Vasily Kobelev[4], Nikolay Kasimov[4], Tatiana Minkina[5], Christopher P. Rüger[1,2], Ralf Zimmermann[1,2]

[1]Joint Mass Spectrometry Centre (JMSC), Chair of Analytical Chemistry, University Rostock, 18059 Rostock, Germany
[2]Department Life, Light & Matter (LLM), University of Rostock, 18059 Rostock, Germany
[3]Skobeltsyn Institute of Nuclear Physics, Lomonosov Moscow State University, Leninskie Gory, 1, 119991 Moscow, Russia
[4]Faculty of Geography, Lomonosov Moscow State University, Leninskie Gory, 1, 119991 Moscow, Russia
[5]Southern Federal University, 344090, Rostov-on-Don, Russia

Correspondence to: Hendryk Czech (hendryk.czech@uni-rostock.de)

**Abstract**

Wildfires in Siberia generate large amounts of aerosols, which may be transported over long distances and pose a threat to the sensitive ecosystem of the Arctic. Particulate matter (PM) of aged wildfire plumes with origin from Yakutia in August 2021 was collected in Nadym city and on Bely Island (both northwest Siberia). A comprehensive analysis of the chemical composition of aerosol particles was conducted by multi-wavelength thermal-optical carbon analyzer (TOCA) coupled to resonance-enhanced multiphoton ionization time-of-flight mass spectrometry (REMPI-TOFMS) as well as by ultra-high resolution Fourier-transform ion cyclotron resonance mass spectrometry (FT-ICR MS). In Nadym city, concentrations of organic carbon (OC) and elemental carbon (EC) were peaking at 100 µg m$^{-3}$ and 40 µg m$^{-3}$, respectively, associated with Angström Absorption Exponents for 405 and 808 nm (AAE$_{405/808}$) between 1.5 and 3.3. The weekly average on Bely Island peaked at 8.9 µg m$^{-3}$ of OC and 0.3 µg m$^{-3}$ of EC, and AAE$_{405/808}$ close to unity. Particularly, ambient aerosol in Nadym city had a distinct biomass burning profile with pyrolysis products from carbohydrates, such as cellulose and hemi-cellulose, as well as lignin and resinoic acids. However, temporarily higher concentrations of 5- and 6-ring polycyclic aromatic hydrocarbons (PAHs), different from the PAH signature of biomass burning, suggests a contribution of regional gas flaring. FT-ICR MS with electrospray ionization (ESI) revealed a complex mixture of highly functionalized compounds, containing up to twenty oxygen atoms, as well as nitrogen- and sulfur-containing moieties. Concentrations of biomass burning markers on Bely Island were substantially lower than in Nadym city, flanked by appearance of unique compounds with higher oxygen content, higher molecular weight and lower aromaticity. Back trajectory analysis and satellite-derived aerosol optical depth suggested long-range transport of aerosol from the center of a Yakutian wildfire plume to Nadym city and the plume periphery to Bely Island. Owing to lower aerosol concentration in the plume periphery than in its center, it is demonstrated how dilution affects the chemical plume composition during atmospheric aging.



## 1 Introduction

The Arctic is a particularly vulnerable region regarding the effects of global warming, with atmospheric temperatures increasing at two to three times the global average rate, which is referred to as Arctic amplification (IPCC, 2013; Schmale et al., 2021). Next to carbon dioxide and other greenhouse gases, particulate matter (PM) emissions transported to the Arctic region contribute to the rapid warming. Black carbon (BC) is emitted by fossil fuel combustion and biomass burning and is linked to the light absorption of the atmosphere as well as of snow or ice surfaces. Long-range transport to the Arctic carries BC and other tracers of anthropogenic and wildfire origin (Bond et al., 2013; Manousakas et al., 2022; Matsui et al., 2022; Moschos et al., 2022; Stohl et al., 2013). Beyond BC, wildfires are a major source of volatile organic compounds (VOCs), primary organic aerosol (POA) and brown carbon (BrC) which can act as a strong absorber of solar radiation at ultraviolet (UV) and visible wavelengths (Farley et al., 2022; Forrister et al., 2015; Fleming et al., 2020).

The frequency and size of wildfire events has increased during recent decades, and the trend is expected to continue due to global warming and the associated rise in extreme weather events (Abatzoglou et al., 2019; Kharuk et al., 2021). As wildfires in the northern regions increase, the long-range transport of wildfire plumes from central Siberia to Arctic regions can get more common, increasing the impact of carbonaceous aerosols in the Arctic on radiative forcing (Calì Quaglia et al., 2022; Yue et al., 2022). Siberian wildfire plumes may even reach densely populated regions in Europe or East Asia (Ikeda and Tanimoto, 2015; Semoutnikova et al., 2018), associated with a significantly increased risk of mortality, respiratory and cardio-vascular diseases of the exposed population (Chen et al., 2021). Although northern boreal regions like Siberia are predicted to be impacted the most by increasing wildfire intensity, studies investigating organic aerosol emissions, especially from Siberia, are scarce (Flannigan et al., 2009).

In addition to individual fuel properties, combustion conditions largely affect the emission composition. Low combustion efficiency with a flameless burning (smoldering) generates aerosols rich in organic matter (OM) (Kalogridis et al., 2018), resembling the composition of the biomass by intense release of phenolic building blocks from lignin, cellulose and hemicellulose (Simoneit, 2002), and containing tar-like BrC with Angström Absorption Exponents (AAE) significantly larger than unity (Chakrabarty et al., 2010). Under flaming conditions, the overall organic aerosol content is reduced and soot-carbon, related to elemental carbon (EC), as well as parent polycyclic aromatic hydrocarbons (PAHs) become more substantial aerosol constituents and the AAE shifts to unity (Popovicheva et al., 2016; Popovicheva et al., 2015).

Atmospheric aging reactions under the influence of, e.g., UV radiation, ozone, $NO_x$ or $SO_x$, transform organic vapors and OM by, e.g., photolysis, hydroxyl radical and nitrate-radical reactions (Peng et al., 2021; Surratt et al., 2008; Forrister et al., 2015). In homogeneous gas phase reactions, organic vapors may be oxidized and condense as secondary organic aerosol (SOA) on existing particles or even form new particles (nucleation). Moreover, heterogeneous reactions between atmospheric oxidants and particle constituents or reaction between individual particle constituents may increase the molecular complexity of primary aerosols in the atmosphere, which is associated with higher functionalization, increase in heteroatom content (O, N, S) and oligomer formation (Schneider et al., 2022; Pardo et al., 2022; Lin et al., 2015; Chacon-Madrid and Donahue, 2011). For biomass burning, atmospheric aging may promote the formation of chromophores and thus BrC, but may also lead to a degradation of chromophores by photochemistry with increasing atmospheric lifetime (Fleming et al., 2020; Forrister et al., 2015).



Wildfires may rapidly increase the level of particulate matter (PM) to a range that atmospheric oxidant
concentrations become insufficient to uniformly process the wildfire aerosol, leading to differences in atmospheric
processing of plume centers and plume periphery (Hodshire et al., 2021). For example, OH radicals, the main
oxidant under photochemical conditions, may be already consumed at the periphery of the plume, which has been
linked to result in different optical plume properties (Palm et al., 2021). Furthermore, other photosensitive
atmospheric oxidants such as $NO_3$-radicals may be protected by the plumes optical depth and become relevant for
the chemistry in the plume center (Decker et al., 2021). Consequently, the product spectrum of atmospheric
processing differs from aging of biomass burning aerosol at typical ambient conditions. Additionally, the net
appearance of OM in wildfire plumes during aging becomes dilution-driven as high near-source aerosol
concentrations release adsorbed and absorbed vapors during atmospheric transport (Palm et al., 2020), on the one
hand counteracting the known significant net increase of OM mass for a wide range of biomass burning aerosol
aging (Ortega et al., 2013), but on the other hand forming secondary organic aerosol (Fang et al., 2021; Li et al.,
2020). As aerosol concentrations at the plume periphery are lower than in the center, they undergo more intense
aging (Hodshire et al., 2021). Quantification of transported wildfire aerosol and its molecular characterization
during aging may improve the understanding of impacts on the sensitive Arctic ecosystem and related effects on
climate.
Siberian wildfires are a major source of climate-relevant species emitted at northern latitudes (Lavoué et al., 2000).
Yakutia in eastern Siberia is known to be prone to large-scale wildfires (Tomshin and Solovyev, 2018) owing to
the combination of hot summers with temperatures up to 40 °C, low humidity in atmosphere and pedosphere, and
the phenomenon of dry thunderstorms, which have been estimated to account for more than a half of the fire
causes. In such, lighting ignites dry biomass while strong wind accelerate the spread of the fire (Narita et al., 2021).
On August 4 of 2021, strong smoke enveloped large areas of western Siberia, namely Yamalo-Nenets Autonomous
Okrug (YNAO) and Khanty-Mansysky autonomous Okrug (KMAO) (NUR24.RU, 2021).
In our study, unprecedented high concentrations of OC and EC were observed in Nadym city and on Bely Island,
located in the North of Western Siberia, during August 2021, according to filter samples collected at both locations.
Backward air mass trajectories and satellite images indicated large-scale wildfires in Yakutia as the main source.
Multi-wavelength thermal-optical carbon analysis with photoionization mass spectrometry and ultra-high
resolution mass spectrometry with complementary ionization techniques confirm the origin of the observed high
OC and EC concentrations and provide a comprehensive chemical characterization of Arctic pollution associated
with aerosol emissions from vast Siberian fires and differences in atmospheric aging of plume center and plume
periphery.
**2 Experimental Section**
**2.1 Sampling sites**
One part of the sampling campaign was carried out in Nadym city from 5 August to 12 August 2021. A total
suspended particle (TSP) sampling system was operated in an area distant from roads and residential sector, with
a flow rate of 70 L min$^{-1}$ and variable duration of 3–12 h to achieve comparable filter loadings of the nine resulting
samples. Quartz fiber filters (QFF, 47 mm, QMA 1851-047, Whatman, USA) were used to collect PM samples,
after 6 h pretreatment at 600°C.



The second sampling system was operated at the pavilion of the research aerosol station "Island Bely". The TSP
inlet was installed approximately 1.5 m above the roof and 4 m above the ground for filter sampling, see for details
elsewhere (Popovicheva et al., 2022). Three QFF were collected by weekly sampling, starting on 31 July 2021 and
ending on 21 August 2021, with a sampling flow rate of 2.3 $m^{-3}$ $h^{-1}$. A more detailed overview of the sampling
parameters as well as an in-depth discussion of the study area and typical PM emissions sources can be found in
the SI section 1.

## 2.2 Air mass transportation

To evaluate the impact of air mass transportation and smoke plume origin, 240 h backward trajectories (BWT)
were generated using the NOAA HYSPLIT model of the Air Resources Laboratory (Stein et al., 2015) and archive
data from the National Center for Environmental Prediction's Global Data Assimilation System with the coordinate
resolution equal to $1° \times 1°$ of latitude and longitude and an input height of 500 m above ground.
Data on the occurrence of fires was obtained from the Fire Information Resource Management System (FIRMS)
operated by the NASA/GSFC Earth Science Data Information System (ESDIS)
(https://firms.modaps.eosdis.nasa.gov/map) based on satellite observations which register open flaming. This work
uses data arrays on the spatial location of fire centers from the Moderate Resolution Imaging Spectroradiometer
(MODIS). The satellite images of smoke plumes in the sampling days were obtained from
https://worldview.earthdata.nasa.gov. Fire activity is shown in 10 days back from a day of BWT analyses (Fig. 2).
The Ozone Mapping and Profiling Suite (OMPS) Aerosol Index information was obtained from Goddard Earth
Sciences Data and Information Service Center (GES DISC) based on satellite instrumentation that measures the
radiance scattered by the limb of the atmosphere. The OMPS Aerosol index is based on the normalized radiance
of the wavelengths 340 and 378.5 nm (Torres, 2019).

## 2.3 Analytical Instrumentation

### 2.3.1 Fourier-Transform ion cyclotron resonance mass spectrometry (FT-ICR MS)

Ultrahigh-resolution FT-ICR-MS measurements were carried out on a SolariX (Bruker Daltonik, Bremen,
Germany) equipped with a 7 T superconducting magnet and an InfinityCell©. A detailed description of the
extraction procedure as well the settings for each ionization technique can be found in the SI section 2. Shortly,
filter extracts were analyzed in positive and negative ionization mode electrospray ionization (ESI), as well as
positive mode atmospheric pressure photoionization (APPI: Kr, 10/10.6 eV) with a direct-infusion ion source setup
(Bruker Daltonik, API Ion Source). For each measurement, 400 Scans were collected in the range of $m/z$ 150–
1,000 with a resulting resolving power >310,000 at $m/z$ 400 and mass accuracy below 1 ppm.

### 2.3.2 Multi-wavelength thermal-optical carbon analysis hyphenated to resonance-enhanced multiphoton ionization time-of-flight mass spectrometry (TOCA-REMPI-TOFMS)

For direct analysis of filter samples, a thermal-optical carbon analyzer (TOCA; Model 2001, DRI, US) hyphenated
to time-of-flight mass spectrometry (TOFMS) with resonance-enhanced multiphoton ionization (REMPI). Organic
and elemental carbon (OC, EC) was determined according to the IMRPOVE_A protocol (Chow et al., 2007). using
laser transmittance (LT) at 635 nm for the separation of pyrolytic OC from EC. In addition to 635 nm, the TOCA
was further retrofitted with six laser diodes, emitting light in the visible UV and near-infrared (NIR) spectral range



at 405, 445, 532, 780, 808 and 980 nm (Chen et al., 2015). The multi-wavelength TOCA was used to determine
the Angström Absorption Exponent (AAE), which is defined for a wavelength pair $\lambda_1/\lambda_2$ by
$$AAE(ATN,\lambda) = -\frac{\ln\frac{ATN_1}{ATN_2}}{\ln\frac{\lambda_2}{\lambda_1}} \tag{1}$$
with ATN being the light attenuation.
In this study, the wavelength pair of 405 and 808 nm was used to calculate the AAE ($AAE_{405/808}$), representing the
exclusive absorption of BC in NIR and lower visible UV range with absorption of both BrC and BC. ATN was
derived from ratio of LT of the untreated filter sample before TOCA to the LT at the end of the TOCA that
refractive particle constituents are still retained. In order to account for filter loading effects on ATN, the empirical
correction from (Chow et al., 2021) was applied. The uncertainty of the AAE determination was derived from
error propagation of the LT measurement, which has a precision of 8 % at 405 and 808 nm (Chen et al., 2015).
Thus, the final uncertainty of the $AAE_{405/808}$ calculation is within ±15 % at 95 % confidence.
A small fraction of evolving particulate matter during IMPROVE_A is bypassed from the carbon quantification
by a modified quartz tubing behind the oven of the TOCA and reaches a REMPI-TOF-MS through a heated transfer
capillary (Grabowsky et al., 2011). REMPI refers to a selective ionization technique for aromatic compounds and
predominantly yields molecular ions (Streibel and Zimmermann, 2014). A more detailed description of REMPI
can be found in the SI section 2.3.
**2.4 Mass spectrometric and statistical data analysis**
External mass calibration of the FTICR-MS was performed using arginine oligomers covering the entire mass
range. An internal calibration of each mass spectrum was performed by characteristic and commonly abundant
peaks from a self-generated calibration list ($CHO_x$, $CHNO_x$ class compounds, manually verified) achieving sub-
ppm mass accuracies. Raw data was peak picked (cut-off: S/N = 6) and exported with Bruker Data Analysis 5.1
(Bruker Daltonik, Bremen, Germany). The exported mass spectra were processed by self-written MATLAB
algorithms and routines combined in a graphical user interface named CERES Processing (Rüger et al., 2017).
After careful investigation and taking into account attribution boundaries from literature, the following restrictions
were deployed for elemental composition assignment in the range of 150–1,000 *m/z*: $C_cH_hN_nO_oS_sNa_{na}$; $5 \leq c \leq 60$,
$5 \leq h \leq 100$, $n \leq 3$, $o \leq 20$, $s \leq 1$ (ESI+: $na \leq 1$) with a maximum error of 1 ppm (Tang et al., 2020; Schneider et
al., 2022). Additional restrictions were applied for the H/C ratio: 0.4–2.4, O/C ratio: 0–1.4 and double bond
equivalents: DBE 0–28. Equations used for the calculation of molecular properties can be found in the SI section

29   2.4.

Principal component analysis (PCA) and hierarchical cluster analysis (HCA) was performed by the MATLAB
R2021b Statistics Toolbox (The MathWorks, Inc., MA). HCA was performed with the following settings:
unweighted average distance (UPGMA), cosine distance metric, max. number of clusters set to 5 and absolute
intensities normalized by power transformation. Prior to PCA, sum parameters were standardized to a mean of
zero and standard deviation of one.
Calculation of relative sum formulae intensities is done by L1-norm (normalization to TIC) of all assigned
elemental compositions of the respective sample.





**3 Results and Discussion**
**3.1. Air mass transportation to Nadym city and Bely Island.**
Figures 2 and S1 present backward trajectories (BWT) for air mass transport at times corresponding to sample
collection in Nadym city and on Bely Island, throughout the sampling campaigns using the HYSPYT simulation
data. We present this analysis first, as it provides useful information for analyses of aerosol chemistry discussed
in later sections. Unprecedented smoke intensity was initially observed in Nadym city on 05/06 August 2021 when
samples N01 and N02 were collected (Table S1). Air masses which arrived on those days to Nadym from NE and
NNE directions passed the wildfire areas in Krasnoyarsky kray (Fig. 1 and S2). As observed from satellite images,
they brought a wide and dense smoke plume covered the YNAO, KMAO, and Yakutia, Krasnoyarsky kray and
Irkutskaya oblast. On 07 August 2021 when sample N03, N04 and N05 were collected, BWT arrived to Nadym
from NE direction, after they passed Yakutia with the largest density of detected fires, then Irkutskaya oblast and
Krasnoyarsky kray, and turned to YNAO (Fig. 2).
On 08 August 2021, the smoke plume area narrowed and localized almost over the territory of Krasnoyarsky kray.
Nadym is observed almost free from smoke when sample N07 and N08 were collected. Finally, air masses changed
their direction and on 09 August 2021 arrived at Nadym from NW and NNW. They brought clean arctic air from
White, Barents and Kara Seas. On 12 August 2021, not even traces of a fire plume are observed over Siberia in
satellite images.
Air mass trajectories and satellite images reveal that the wildfire plume that has strongly impacted Nadym also
reached further north into the Arctic and brought deep smoke into the Bely Island during the same period from 05
to 07 August 2021. In contrast to the plume arriving at Nadym, the northern part of the evolving wildfire plume
was first transported north from its origin for several days, and moved around over the Arctic Ocean, before a
change of wind direction transported the plume westward. The OMPS Aerosol Index (Fig. S2) suggests that the
periphery (lower OMPS Aerosol Index, yellow) of the Yakutian wildfire plume was transported to Bely Island in
contrast to the plume center aerosol transported to Nadym (higher OMPS Aerosol Index, red). This may have led
to a gradient in photochemical processing of the plume, i.e., a lower extent of atmospheric processing by OH
radicals, with the northern section containing more atmospherically aged aerosol and the southern section more
fresh wildfire emissions, which were picked up on the way westward.
**3.2 Carbonaceous aerosol in Nadym city and on Bely Island**
Long-range transport of a wildfire plume from Yakutia caused high carbon concentrations in Nadym ambient PM
from 05 to 07 August 2021 (samples N01–N07) (Fig.3). For sample N04, concentrations of organic carbon (OC)
and elemental carbon (EC) approached a maximum of 100 and 40 µg m$^{-3}$, respectively (Table S1). On 08 August
2021, OC and EC dropped to 5.4 and 1.7 µg m$^{-3}$ for sample N08, respectively. On 09 August 2021 transport of
clean Arctic air masses decreased OC and EC down to 0.7 µg m$^{-3}$ of OC and 0.1 µg m$^{-3}$ of EC for N09, while on
12 August 2021 3.7 µg m$^{-3}$ of OC and 1.2 µg m$^{-3}$ of EC was obtained for N10, respectively (Fig. 3). Regarding the
distribution of the individual carbon fractions, no significant difference between the samples could be determined
apart from a significantly higher contribution of EC2 in sample N10 (two-sided Grubbs test, significance level of
0.05), collected when smoke is almost disappeared (Table S1). EC2 has been associated with soot particles from
internal combustion engines and indicates that local emission, e.g., from road traffic, contributed significantly the



carbonaceous aerosol. A generally substantial contribution of pyrolytic OC ($OC_{pyro}$) is an indication for the
presence of biomass burning (BB) and secondary organic aerosol (SOA)  (Grabowsky et al., 2011; Cheng et al.,

3     2011).

In the week from 31 July 2021 to 06 August 2021 (sample B01, Bely Island) when the wildfire plume arrived in
Nadym city, concentrations of OC and EC at Bely Island reached the highest weekly averages of 8.9 and 0.3 µg m$^{-}$
$^3$, respectively, and declined in two subsequent weeks to 3.9 (B02) and 0.5 µg m$^{-3}$ (B03) of OC and 0.3 (B02) and
<0.05 µg m$^{-3}$ (B03) of EC (Table S1). PM of Bely Island was substantially affected by the wildfire emissions in
Yakutia (Fig. 2), as was Nadym, but lower concentrations of carbonaceous aerosol particles were observed as only
the periphery of the plume with lower aerosol concentrations was transported to Bely Island (Fig. S2).
The wavelength dependence of light absorption in the UV/vis to near-infrared range is described by the AAE.
While BC shows an AAE close to unity, BrC has stronger increase in absorption towards lower wavelengths in
the visible and UV range with AAE significantly larger than unity (Andreae and Gelencsér, 2006). Regarding
biomass burning, spectral absorption obtained throughout the near-ultraviolet to near-infrared spectral region and
high Angstrom absorption exponents (AAE) up to 4.4 are were found for smoke from smoldering combustion of
pine debris in the wavelength regions from 370 to 670 nm. In contrast, open flaming smoke from pine combustion
shows low AAE around 1, also typical for high-temperature fossil fuel combustion while mixed fires emit particles
absorbing light with the intermediate AAE characteristics (Popovicheva and Kozlov, 2020).
For N01–N05, AAE for the wavelength pair of 405 and 808 nm ($AAE_{405/808}$) were observed in the range of 1.5 to
3.3 (Fig.3), showing the presence significant amounts of BrC. Although there was no distinct relation of $AAE_{405/808}$
to concentrations of OC or EC, a moderate correlation coefficient of 0.69 was obtained between $AAE_{405/808}$ and
the ratio of OC to EC. This correlation indicates the contribution of aerosol from smoldering biomass burning,
which is known to yield in lower amounts of EC and BC, but higher release of BrC than flaming biomass burning
(Cheng et al., 2011; Chen et al., 2006; Popovicheva et al., 2016). For N09, a moderately high $AAE_{405/808}$ of 1.9
was observed, which may be caused by still significant relative biomass burning contribution to overall low OC
and EC concentrations. After the wildfire plume left Nadym city, the lowest $AAE_{405/808}$ of 1.3 (N010) among all
samples from Nadym city was observed, pointing towards dominating aerosol emissions from fossil fuel
combustion associated with a higher contribution of BC (Helin et al., 2021).
At Bely Island, the weekly average of $AAE_{405/808}$ of 1.0 for sample B01 (31 July to 06 August 2021) was not
affected by BrC in the wildfire plume during highest concentrations of OC and EC. In the subsequent two weeks
with lower OC and EC concentrations, $AAE_{405/808}$ of 1.2 were obtained, which are, however, not significantly
different from 1 considering the associated measurement uncertainty of ±0.5. Since lower aerosol concentrations
from the periphery of the Yakutian wildfire plume reached Bely Island, it can be assumed that the wildfire aerosol
has been more intensively processed during atmospheric transportation. Atmospheric ageing may form BrC (Al-
Abadleh, 2021), but particularly toward longer photochemical age, the phenomenon of photobleaching and
whitening of BrC by atmospheric oxidants becomes dominant, decomposing chromophores and consequently
decreasing AAE (Fang et al., 2021; Schnitzler et al., 2022) caused by a higher ratio of atmospheric oxidant to BB
aerosol.
In Salekhard city approximately 350 km west from Nadym and 800 km south from Bely Island, eBC was
continuously measured during summer 2018 with an average concentration of (350±120) ng m$^{-3}$ (Popovicheva et
al., 2020). Although EC and BC are determined by different measurement principles, they are highly correlated,
appear in a similar concentration range and essentially represent graphitized carbon (Watson and Chow, 2002;



Andreae and Gelencsér, 2006; Chow et al., 2021), thus enabling inter-comparison estimates. The observed
concentrations of EC in Nadym city during 05 to 07 August 2021 exceeded the BC concentrations in Salekhard
city by two orders of magnitude. Furthermore, even compared to the highest average levels for July and August
from 2003 to 2017 of fine particulate matter (PM2.5), primary OM, and BC caused by transported wildfire aerosol
to YNAO (Yasunari et al., 2021), carbonaceous aerosol concentration observed in Nadym city in our study remain
significantly high. At Bely Island, ten times higher concentrations of OC and EC were observed compared to
averages of organic aerosol and BC during summertime (Moschos et al., 2022; Popovicheva et al., 2022).
Therefore, our results give rise to unprecedented high concentrations of long-range transported wildfire aerosol to
the Arctic with different aging conditions between the two sampling sites.
**3.3 PM bulk composition by APPI and ESI FT-ICR-MS**
All filter samples extracts analyzed by FT-ICR MS were measured with three atmospheric pressure ionization
techniques: ESI+, ESI- and APPI (+). A detailed discussion of the selectivity and sensitivity of each ionization
technique can be found in the SI section 3.
Principal component analysis (PCA) based on FT-ICR MS average elemental compositions as well as other sum
parameters, e.g., DBE, AI, H/C and O/C, (Table S3, Fig. S4) shows a clear grouping of samples N01 to N02 and
N03 to N05 with a divergence of N07 and strong separation of N08 to N10. For chemical comparison of the FT-
ICR MS data, samples are combined and classified based on the PCA results, previously discussed air mass
trajectories, as well as the EC and OC concentrations. For Nadym, samples N01 to N05 are combined to form one
dataset representing the strongest wildfire plume impact causing high OC and EC concentrations (Fig. 3, Fig. S1).
In contrast, samples N08 to N10 had the lowest concentrations of OC and EC.  N09 was chosen as a reference for
the absence of a wildfire impact and termed as "ambient aerosol" for Nadym city. The apparently different
chemical composition of sample N07, indicating the influence of regional gas flaring, is separately addressed in
section 3.6. For Bely Island, sample B01 is selected for comparison with the Nadym city dataset, as it represents
the strongest wildfire impact at this location, and additionally covers a similar time period (21-07-31 to 07-08-21)
as the Nadym wildfire plume impacted samples N01 to N05 (21-08-05 to 21-08-07). Samples B02 and B03
represent declining wildfire aerosol influence and ambient aerosol at Bely Island, respectively
For the wildfire plume impacted samples (N01–N05), there are 1108 compounds common between all ionization
techniques (Fig. S3), but most compounds are uniquely identified by a single ionization technique. Notably, ESI+
and APPI share the highest number of common elemental compositions (1361), while ESI+ and ESI- share the
lowest number (553), of two ionization techniques. In general, almost every compound detected in any ionization
mode is part of a homologous series spanning over several $CH_2$ units in the range of often 20 or more carbon atoms
(Fig. S5). Other homologous series, including e.g. methoxy groups, are also observed. The Van Krevelen diagrams
(Fig. 4B) show complex fingerprints, with changing highest intensity regions for each ionization technique. The
lipid region (H/C > 2, low O/C, low aromaticity, low carbon oxidation state) is highly abundant, as well as lignin-
like structures (medium H/C, medium O/C, AI < 0.5 and $OS_c$ 0 to -1). In ESI, highly oxidized compounds with
sugar-like structures (high O/C, low aromaticity, high $OS_C$) as well as highly oxidized molecules with high
unsaturation (HU-HOMs) are observed additionally. This highlights the need for different ionization techniques
to achieve a broad coverage of the chemical space.



Saturation vapor pressure is a parameter used to characterize SOA (Donahue et al., 2011; Donahue et al., 2012).
Observed compounds in the wildfire plume show low to very low volatility, as result of high oxygen-contents, and
other hetero elements (N, S), gained by atmospheric aging during long-range transport or maintained due to
reduced photochemical processing. The majority of compounds is found in the low to ultra-low volatility area, but
there is a difference when comparing individual compounds classes.
**3.4 Compound class characterization of PM in Nadym city**
**3.4.1 CHO**
Mass spectra of identified elemental compositions (Fig. 4A) show a broad distribution over the whole mass range
(approx. 200-800 Da) of the highest abundant compound class. The relative intensity distribution of oxygen
number shows a high degree of oxidation (Fig. S11), with ESI- showing the highest average oxygen number of
9.6 oxygen atoms per molecule (Table S3). This is an indicator for acidic functional groups, e.g., hydroxyl (R-
OH) or carboxylic acids (R-COOH), which are efficiently ionized by ESI- and lead to the sensitive detection of
highly oxidized compounds. High oxygen-content is also reflected in a low volatility, which is observed for the
wildfire plume impacted samples (Fig. 5).
Van Krevelen (VK) plots show a wide distribution over several structure regions, including the lipid-, phenol-,
and carbohydrate-like regions, as well as oxidized aromatic compounds. All of them are known from literature to
be regions of products of biomass burning and atmospheric aging of BB emissions. They are not found in PM
samples from Nadym (N08–N10) collected after the plume had passed (Fig. S7). Especially ESI- shows intense
distribution of signals in the center of the VK plot (medium H/C and O/C), while in each polarity the lipid region
of low-oxidized compounds is highly populated. Both regions show no high abundant peaks after 08 August 2021,
indicating no significant contribution of fresh biomass burning emissions to ambient PM. Typical elemental
compositions of BB markers are found in the wildfire plume affected PM samples from Nadym city: levoglucosan
and its isomeric anhydrosugars, resin acids, methoxy-phenols and lipids. Figure S6 shows semi-quantitative time
trends (normalized to sampling volume) of six biomass burning marker elemental compositions, including
elemental compositions of known cellulose and lignin degradation products. The thermal degradation of cellulose
structures leads to the formation of the anhydrosugars levoglucosan as well as minor amounts of its isomers
galactosan and mannosan. Coniferyl alcohol is a gymnosperm lignin degradation product found, e.g., in pine wood
smoke (Simoneit et al., 1993). 7-Oxodehydroabietic acid is emitted from burning of Gymnosperm plants, e.g.,
scots pine and larch, which are the dominant forest types in Yakutia (Kharuk et al., 2021). Nonacosene and
nonacosanol are biomarkers emitted from higher plant waxes (Simoneit and Elias, 2000). A clear, similar trend is
visible for each marker compound, as the abundance increased with a maximum on 07 August 2021 (N04),
followed by a slow decrease to zero towards the end of the observed wildfire plume impact (09 August 2021).
Finally, some peaks are detected in the lower VK space with H/C<0.7 and 0.2<O/C<1.5, particularly in mass
spectra of APPI and ESI-. This has been assigned to HU-HOM, which are produced from the photooxidation of
larger PAHs on soot particles, thus indicating heterogeneous processing of wildfire aerosol particles. The
formation of HU-HOM changes the polarity of soot particles from hydrophobic to hydrophilic and therefore affect
the uptake of water and cloud formation (Li et al., 2022).
The maximum carbonyl ratio (MCR) is a tool to predict the possible maximum of carbonyl groups in molecules,
based on the identification of elemental compositions (Zhang et al., 2021c). The applied ionization techniques





display differences in the distribution of relative intensity into the five defined MCR areas (Table S4). ESI+ data
indicates that there are manifold combinations of oxygen-containing functional groups including not only
carbonyl, but also single-bound oxygen as found in hydroxyl (R-OH), hydroperoxyl (R-OOH) or ether (R-O-R) is
part of this organic aerosol, as there is a relatively even distribution of compounds for each MCR limit. Comparing
the wildfire plume affected samples (N01 to N07) with after-plume samples (N08-N10), it is apparent that the
distribution of the MCR of samples N08-N10 is shifted to lower MCR, indicating a stronger influence of products
from oxidative atmospheric aging. This trend is observed for all applied ionization techniques and may be
explained by the lower concentrations of VOCs and thus lower reactivity towards atmospheric oxidants, which
lead to more oxidized aging products due to a higher ratio of oxidants to organic carbon compared to dense wildfire
plumes.

## 12    3.4.2 CHNO

Wildfires, particularly under flaming conditions, are known to emit reactive nitrogen species like nitrogen oxides
($NO_x$) and nitrous acid (HONO) (Peng et al., 2021; Lindaas et al., 2021). These short-lived compounds may
consequently be converted to long-lived peroxyacetyl nitrate (PAN), nitric acid ($HNO_3$) organic nitrates or other
organic nitrogen-containing compounds (Peng et al., 2021).
PAN and $NO_X$. take part in atmospheric aging reactions, forming organic aerosol molecules containing the
elements oxygen and nitrogen. Most common functional groups are nitrate, nitro and nitrooxy moieties. Also
nitrogen-containing compounds are a relevant portion of BrC due to their potential light-absorbing properties,
especially in combination with aromatic ring-systems (e.g. nitro-phenol derivatives) (Fleming et al., 2020).
The CHON class is the second most intensive, but most frequently detected, compound class in PM of Nadym city
affected by wildfires, with molecules containing commonly one or two nitrogen atoms. Most compounds of the
CHON class show a high degree of oxidation, comparable to the CHO class, with the same shift to lower oxygen-
number for ESI+ (Fig. S11). Notably, the oxygen distribution of the ESI- data shows the maximum at 9 oxygen
atoms, independent of the nitrogen number. An O/N ratio $\geq$ 3 is the limit for the potential presence of an organic
nitrate group, but as the elemental compositions show a high degree of oxidation (O/N >> 3), the detection of
hydrocarbons with only nitrate groups is rather unlikely.
Nitroorganics are a relevant species regarding their light-absorption properties and their role in BrC (Salvador et
al., 2021). More likely, the majority of compounds detected by ESI and APPI contain more than one functional
group, and therefore no clear identifications solely based on the O/N ratio is possible. On the other hand,
compounds with an O/N < 3 can be identified as potential nitro compounds, which are detected here frequently by
ESI+ (998) and APPI (390).
ESI+ is also detecting a number of compounds with more or equal amounts of nitrogen compared to oxygen. These
compounds could be assigned to alkaloid-like structures or other moieties that include nitrogen-containing
heterocyclic compounds (Laskin et al., 2009), which are preferably released under low-temperature and oxygen-
poor combustion conditions (Ren and Zhao, 2015).




### 3.4.3 Sulfur-containing compounds (CHOS/CHNOS)

Sulfur is, to some degree, a part of biomass, e.g., as disulfide bonds or in certain amino acids, and compounds containing sulfur are therefore emitted by biomass burning. Second, e.g. organosulfates are markers for secondary organic aerosol formation with reactions of precursors with anthropogenic pollutants, including sulfates, dimethyl sulfide and other sulfur-containing nucleophiles like $SO_2$ (He et al., 2014; Ye et al., 2021). Additionally, marine biogenic emissions of reduced sulfur compounds are a major source of dimethyl sulfide (DMS), carbon disulfide ($CS_2$) and their oxidation product carbonyl sulfide (OCS) (Qu et al., 2021; Lennartz et al., 2017). The presence of nitrogen in the precursor, as well as the high abundance of $NO_x$ may also lead to the formation of nitrooxy-organosulfates (He et al., 2014).

CHOS compounds, observed almost exclusively by ESI-, display four to eleven oxygen atoms (Fig. S10) as well as low DBE values, indicating long-chain aliphatic structures with one sulfate group and additional oxygen-containing groups. Identical sum formulae have been previously observed in aerosol samples from Chinese megacities (Wang et al., 2016), as well as the ozonolysis of isoprene SOA in the presence of acidic sulfate aerosol (Riva et al., 2016). For example, the $CHO_6S_1$ class is dominated by one homologous series (DBE = 2) reaching from $C_5H_{10}O_6S_1$ (198.01981 Da) to $C_{35}H_{64}O_6S_1$ (612.44236 Da), including several sum formulae known from literature, with a proposed terpene origin and the structure of an aliphatic chain with one sulfate, hydroxyl and carbonyl group each (Riva et al., 2016). Only few compounds display higher DBE values (DBE = 7), indicating the minor abundance of aromatic precursors. Some of these CHOS compounds are also found in the Nadym city PM samples after the plume had passed (N08–N10), but less often and only at higher molecular weights. This may be explained by an independent source of acidic sulfate aerosols e.g., originating from sea spray or marine emission sources (Zhang et al., 2021a), leading to the formation of the observed compounds by reactions with biogenic emitted terpenes e.g. via the epoxide pathway, as well as the presence of other sulfur-containing functional groups, apart from sulfates, like sulfides or thiophenes.

Nitrooxy-organosulfates are also observed in the wildfire plume impacted samples (N01–N05), but with a lower relative intensity compared to CHOS compounds. These compounds are, for example, a product of the combination of two aging reactions adding a nitrate (or nitrooxy) and a sulfate groups to one molecule. The sum formulae of one exemplary compound $C_{10}H_{17}N_1O_7S_1$ is suggested to contain one nitrate and one sulfate group, within several isomers, and was also identified in aerosol samples from Shanghai and nighttime oxidation experiments of monoterpenes under acidic conditions (Wang et al., 2016; Surratt et al., 2008). In addition, the sulfur-content in CHNOS compounds can also be in the form of sulfides, sulfones, sulfur in ring systems, or other functional groups (Ditto et al., 2021). Detected CHNOS compounds in the main plume are almost exclusively comprised of ELVOC or ULVOC. Similar results for low volatility CHNOS compounds in aerosol over agricultural fields were linked to a biogenic origin with formation based on sulfate addition to epoxide CHON precursors (Vandergrift et al., 2022).

### 3.3.4 Reduced compounds (CH/CHN)

Pure hydrocarbons are only detected by APPI, due to the poor ionization efficiency of non-polar molecules with ESI, as well as the good ionization efficiency of photoionization for most hydrocarbons (Kauppila et al., 2017). It is expected that hydrocarbons are not found in high abundance in an aged, long-range transported wildfire plume, as oxidation reactions are leading to the transformation of pure hydrocarbons or non-oxygen-containing species.



Nevertheless, due to the plume optical thickness protecting from photolysis and general oxidant deficit, some
hydrocarbons are observed, including an extensive homologous series of alkenes from $C_{15}$ to $C_{33}$. (DBE = 1) as
well as higher aromatic structures potentially identified as alkylated (aromatic) ring systems (DBE = 6–9).
Reduced nitrogen compounds are only found in ESI+ which indicates amine, pyridine or other basic aromatic
nitrogen moieties in the detected molecules. They are found in the moderate DBE range of 4–8, so the occurrence
of aromatic structures is possible. Proposed alkaloid structures, in the same DBE range, also containing two or
more nitrogen-atoms in one molecule, are a known product of incomplete biomass burning, e.g. due to high
concentrations in ponderosa pine foliage and thermal stability of these compounds (Laskin et al., 2009).
**3.5 Carbohydrate, lignin and resinoic acid thermal degradation products in PM of Nadym city**
The REMPI mass spectra from OC fractions OC1 and OC2 were combined to OC1-2, which is dominated by
thermal desorption, and OC3 and OC4 to OC3-4, showing a shift towards smaller molecular masses due to thermal
decomposition of larger chemical structures as a complementary approach to FT-ICR MS. As discussed in section
3.4.1 about CHO species, BB releases monomers from the decomposition of the biopolymers cellulose,
hemicellulose and lignin, which are commonly used as BB marker. However, as the detection of lignans, such as
tetrahydro-3,4-divannilylfuran suggests that also phenolic dimers and larger thermal lignin fragments are emitted
(Oros and Simoneit, 2001). Those larger fragments resist the temperatures in OC1-2, but decompose *in situ* to
monomers in OC3-4, which indirectly allows BB identification. Previous studies described REMPI mass spectra
from the pyrolysis of cellulose, softwood-derived lignin (Grabowsky et al., 2011), several types of biomass (Fendt
et al., 2012; Fendt et al., 2013) and SOA from ozonolysis of β-pinene (Diab et al., 2015). Due to the highly oxidized
nature of cellulose and SOA, thermal decomposition is similar and complicates the assignment of the mass
spectrometric pattern.
All samples which are strongly affected by the wildfire plume exhibit a minimum uncentered correlation
coefficient of 0.94. This is in agreement with the similar air mass trajectories to Nadym arriving from 05 to 07
August 2021, while the similarity to the background days is only 0.71 on average. During pyrolysis of low-volatile
oxygenated compounds, oxygenated aromatic species are formed, such as phenol (*m/z* 94), cresol (*m/z* 108),
benzofuran (*m/z* 118) and naphthols (*m/z* 144, *m/z* 158). Hence, most abundant peaks mainly belong to *m/z* found
in OC3 of cellulose and SOA. The pyrolysis yield of lignin-derived species depends on the type of biomass, either
gymnosperm, angiosperm or grasses. Although central Yakutia is largely covered by coniferous plants like larch
and pine (Kharuk et al., 2021), *m/z* of syringol-type methoxyphenols (syringaldehyde, *m/z* 182; allylsyringol,
*m/z* 194;sinapyaldehyde, *m/z* 208) from angiosperm can be detected next to guaiacol-type methoxyphenols from
gymnosperm (guaiacol, *m/z* 124; methylguaiacol, *m/z* 138; eugenol, *m/z* 164; coniferyl alcohol, *m/z* 180).
Moreover, hydroxyphenols from the decomposition of less lignified plants or plant material (B. Simoneit, 2002)
are present in the REMPI mass spectra (vinylphenol, *m/z* 120; dimethylphenol, *m/z* 122, cinnamic alcohol,
*m/z* 134; anisaldehyde, *m/z* 136).
In addition to monoaromatic compounds, polycyclic aromatic hydrocarbons and their derivatives are present in
the sample or formed in situ. Retene or 1-methyl-7-isopropyl phenanthrene refers to an established marker for the
combustion of coniferous biomass (Ramdahl, 1983). It is formed from the thermal degradation of diterpenoids
with abietane skeleton. For pine and Siberian larch, most abundant diterpenoids are abietic acid and isopimaric
acid as well as levopimaric and palustric acid, respectively (Bardyshev et al., 1970). Their thermal degradation



depends on the individual combustion conditions and involves successive dehydrogenation, dealkylation and
decarboxylation, finally resulting in an aromatization of one to three six-rings. Consequently, a broad product
spectrum is obtained with retene as the most likely reaction product (Standley and Simoneit, 1994; Marchand-
Geneste and Carpy, 2003), which has been recently investigated with TOCA-REMPI-TOFMS for spruce logwood
and brown coal briquette combustion (Martens et al., 2023). Despite being an alkylated PAH, retene gives a
relatively low yield of the molecular ion and partially fragment during REMPI causing dehydrogenation and
demethylation to $m/z$ 232 and $m/z$ 219, respectively. In OC3-4, retene ($m/z$ 234) is formed in situ from earlier
thermal degradation products of diterpenoids, such as dehydroabietic acid or simonellite, and highly correlates
with the total OC concentrations in Nadym city (Fig. 6B), giving evidence for a dominating OC source of burning
coniferous biomass. In OC1-2, containing retene which is truly in the sample, also a correlation can be observed.
However, intensities of $m/z$ 234 in OC1-2 are apparently lower than for OC3-4. Primary retene ($RET_{prim}$), which
has been formed during combustion and is detected in OC1-2, is associated with more efficient combustion than
pyrolytic retene ($RET_{pyr}$), which is formed from the pyrolysis of diterpenoids and their alteration products and
detected in OC3-4. Hence, the ratio of primary retene to total retene ($RET_{tot}$) in OC may provide a metric for the
combustion efficiency of coniferous biomass, similar to the modified combustion efficiency based on CO and $CO_2$
(Yokelson et al., 1996), which is further evaluated in section 4 of the SI. According to $RET_{prim}/RET_{tot}$ close to
zero, samples N01–N02 (06 August 2021) contain biomass burning aerosol e.g. originating from fires at
smoldering condition (Fig. S6). From 07 August 2021 to the morning of 08 August 2021 (N03–N07), the fires
became more intense and turned over to more flaming conditions, suggested by increased OC and EC
concentrations, lower ratios OC-to-EC being typical for higher combustion efficiency, and $RET_{prim}/RET_{tot}$ between
0.15 and 0.26; on these days, the main plume by means of highest aerosol concentrations arrived Nadym city.
During the reference days with typical background concentrations, one to three orders of magnitude lower
intensities for $RET_{pyr}$ and no $RET_{prim}$ were detected, indicating low biomass burning activity. The same approach
could be principally used for lignin-derived methoxyphenols, such as vinylguaiacol ($m/z$ 150) (Fig. 6D), but it will
be more affected by the dilution because of their higher volatility. Beyond methoxyphenols and retene, REMPI is
particularly sensitive toward PAH (Streibel and Zimmermann, 2014). In OC3-4, two- to four-ring PAH ($m/z$ 128,
$m/z$ 154, $m/z$ 166, $m/z$ 178, $m/z$ 202, $m/z$ 228) are formed by pyrolysis, but larger PAH with five or more aromatic
rings ($m/z$ 252, $m/z$ 276, $m/z$ 278, $m/z$ 300, $m/z$ 302) are divided between OC1-2 and OC3-4 (Diab et al., 2015).
Also for $m/z$ 276, representing six-ring PAH like benzo[g,h,i]perylene or indeno[c,d]pyrene, a good correlation
can be observed with OC (Fig. 6F). However, in OC vs $m/z$ 276 of OC1-2 an outlier emerged, belonging to sample
N07. The impact of regional gas flaring and associated emission of PAH is separately discussed in the section 3.6.
To estimate the influence of the wildfire plume on ambient particle composition and concentration, a REMPI
spectrum of OC3-4 from sample N09 (09 August 2021) with OC and EC concentration below 1 µg m$^{-3}$, typical for
the Arctic region average (Yasunari et al., 2021), is shown (Fig. 6E). More than two orders of magnitude lower
intensities were found for lignin-related $m/z$ and one order of magnitude lower intensities for aromatics formed
from pyrolysis of low-volatile oxygenated compounds. Due to the absence of $RET_{pyr}$, we conclude that those ions
belong to pyrolysis of SOA rather than fragments of carbohydrates. The base peak at $m/z$ 117, however, belong to
indole, which has a 32-40 higher photoionization cross section at 266 nm than toluene (Gehm et al., 2018) and
results from the pyrolysis of bioaerosol components, such as proteins (Fuentes et al., 2010), which emphasizes the
higher contribution of the natural background to the PM composition in Nadym.



**3.6 Impact of Gas Flaring on PM composition in Nadym city**
In addition to long-range transported wildfire emissions, gas flaring from open excess-gas burning at oil and gas
fields is one of the major sources of black carbon (BC) emissions in the Siberian Arctic (Popovicheva et al., 2022;
Stohl et al., 2013; Popovicheva et al., 2017). Sample N07 was not apparent in OC and EC compared to the other
wildfire plume impacted samples N05 and N08 collected on the days before and afterward, but differed in chemical
composition. Backward air mass trajectories for N07 sampling show the transport of air masses through the region
south of Nadym, where many oil and gas fields are located (Fig. S1). A unique pattern of compounds is observed
in both, APPI-FT-ICR MS and TOCA-REMPI MS, with each ionization technique being sensitive for the detection
of aromatic hydrocarbon compounds, due to the favorable photoionization properties of aromatic ring structures
(Gehm et al., 2018; Huba et al., 2016). When excess gaseous organic compounds are burned in a gas flare,
incomplete combustion can lead, among others, to the formation of aromatic hydrocarbons in a large range of
molecular size, from benzene to aromatic soot precursors with large condensed ring structures in various structural
combinations and sizes (Slavinskaya and Frank, 2009; Zhang et al., 2021b; Senkan, 1996). Figure 7A shows a
comparison of averaged samples impacted by the wildfire plume (N01–N05) and sample N07 APPI data,
highlighting unique, condensed aromatic compounds (AI > 0.67), with up to 40 carbon atoms, exclusively
observed in N07. Due to their pyrogenic formation, no pronounced alkylation is observed in these compounds,
which is a key factor for the differentiation to condensed aromatic structures e.g. found in petroleum (Fig. 7B). By
calculating the slope of the planar limit and the ratio of core to methylated species intensity ($C_0/C_0+C_1$) in the
sample N07 (Fig. 7B, Table S5), the pyrogenic origin (slope: 0.81, $C_0/C_0+C_1$: 0.7–0.8) is confirmed (Yunker et al.,
2002; Cho et al., 2011). A slope of the planar limit from 0.75 to 1 indicates the addition of benzene rings linearly
or nonlinearly to a core structure, and a maximum of intensity for each DBE value at $C_0$ is associated with
combustion emissions (Laflamme and Hites, 1978).
As pointed out in section 3.5, markers of BB in the TOCA-REMPI mass spectrum of OC34 strongly correlates
with the total content of OC. However, for $m/z$ of PAH, such as $m/z$ 276 representing six-ring PAHs, the sample
N07 of 08 August 2021 deviates from this correlation (Fig. 6F). In the REMPI mass spectrum of OC12, minor
fragments at $m/z$ 118, 132 and 146 from low-temperature pyrolysis of carbohydrates and SOA are visible, but
larger parent PAH dominate the mass spectrum of N07 in contrast to wildfire plume sample N03 (Fig. S8) and
supports the findings from APPI-FT-ICR MS.
Despite apparent differences in air mass trajectories to N05 or N08, sample N07 containing PM from long-range
transported wildfire plume, the mass spectrometric analyses suggest the presence of a high-temperature
combustion aerosol, such as from gas flaring, which was added to the chemical PM signature of biomass burning
and aged organic aerosol. Both wildfire and gas flaring can be regarded as the most significant contributors to PM
in northern Siberia.

**3.7 Chemical characterization of PM from Bely Island**
FT-ICR MS data of wildfire plume affect PM on Bely Island, collected during the same period of high PM levels
in Nadym city (sample B01, 31 July to 06 August 2021), shows a similar or even higher complexity then the PM
in Nadym city as well as partly different chemical composition. Figure 8 shows mass spectra of the three most
abundant compound classes in the range of $m/z$ 150–800 in APPI and ESI, with APPI showing approximately the
same number of identified sum formulae (5231), ESI- showing a lower number (3553) and ESI+ showing even



more identified formulae (7361) in PM on Bely Island (B01) compared to the composition of PM in Nadym city
averaged over 05 to 08 August 2021 (N01 to N07). When comparing the spectra of Fig. 8 with data from Nadym
(Fig. S10), differences in the chemical composition become apparent.
The most pronounced region in the APPI spectrum of Nadym PM samples affected by wildfire plume (N01–N07)
is *m/z* 400–500 containing high intensities of CHO1–CHO4 compounds with moderate aromaticity (DBE 5–10).
This section is less abundant in the mass spectrum from PM of Bely Island (B01), but the lower *m/z* region (200–
350 Da), including CHO5–CHO7, compounds is more pronounced. The same shift to higher degree of oxidation
can be observed in the ESI+ data of the PM samples from Bely Island. The CHO8–CHO14 compounds compile
the broad signal distribution of the CHO class at *m/z* 400–800 in the PM samples from Bely Island, while the ESI+
spectrum of Nadym PM samples is characterized by single intense signals of e.g. marker compound masses like
levoglucosan, which is not identified on Bely Island, and a more equal distribution of the remaining majority of
signals.
The lower abundance of biomass burning marker compounds in PM samples from Bely Island could be attributed
to more pronounced aging and SOA formation of the air masses reaching the Arctic region, reducing the amount
of primary biomass burning markers.
In the REMPI mass spectra of OC3-4, BB-related thermal fragments discussed in section 3.5, are clearly visible
for the weekly samples from 31 July to 07 August 2021 (B01) and 07 to 14 August 2021 (B02), respectively (Fig.
S9A, B). However, compared to a main plume sample from Nadym (N05) (Fig. S9D), these samples from Bely
show distinct higher *m/z* in OC3-4, possibly due to the formation of larger, chemically different or more stable
structures. The ratio of RET$_{pyr}$ to OC in B01 and B02 is 30 to 50% lower than in sample N05, but still confirms
the significant contribution of BB aerosol. In the sample after the plume event from 14 to 21 August 2021 (B03),
both overall and BB-related thermal fragments disappeared in the REMPI mass spectrum, while N-containing
thermal fragments from the degradation of bioaerosol components, such as from proteins, increased in relative
abundance at *m/z* 117 (indole) and *m/z* 131 (methyl-indole) (Fig. S9C).
When comparing sum parameters determined from the elemental composition assignment of both datasets (Table
1, Tables S3, S6, S7), a trend resulting from an increased photochemical age is visible. All ionization techniques
show that the samples collected on Bely Island are more oxidized and less aromatic with a higher saturation vapor
pressure, as well as higher average O/C and O/N ratios. The relative CHO intensity is increased, while the other
relative intensities are decreased. Also, AAE values are decreased, indicating degradation of chromophores by
photobleaching (Liu et al., 2021). This behavior indicates more intense atmospheric aging, especially by reactions
adding oxygen to the organic aerosol molecules, but not nitrogen or sulfur. This is contrary to observations made
for the evolution of emissions from a boreal forest fire in Lac La Loche (Canada), where an increase of nitrogen-
and also sulfur-containing compounds was observed (Ditto et al., 2021). The lower concentrations from the
wildfire plume periphery lead to a higher ratio of atmospheric oxidants to reactive aerosol constituents.
Furthermore, the more remote location of Bely Island compared to Nadym city determines a lower mixing with
reactive gases from anthropogenic emissions (e.g., NO$_x$ and SO$_x$). Therefore, the PM composition in Bely Island
shows a picture of more intensively oxidized organic matter with a lower content of N- and S-containing
compounds compared to PM from Nadym city.
For a comprehensive insight into the differences in PM composition in Nadym and on Bely Island, the intersect of
both datasets in each ionization technique is determined (Fig. 9A). The compound class distribution of each section
gives an overview over the chemical composition unique for each sampling location. A large fraction of sum



formulae is present in both datasets, which is explained by the almost identical source of the PM emissions.
Nevertheless, a high number of compounds is uniquely abundant in one of the two datasets. The *m/z* unique for
Nadym city PM includes a disproportionate high number of nitrogen-containing compounds (CHNO), while the
PM sample from Bely Island contains more CHO compounds (except for APPI) relative to CHON compounds.
Van-Krevelen plots of all applied ionization techniques reveal a clearly visible difference in the fingerprint of
unique organic compounds (Fig. 9). Peaks only detected in the Nadym city PM are in the low to medium O/C
range, with a H/C ratio larger 1.2 and an average $OS_C < -1$. On the other hand, unique peaks in PM of Bely Island
shows a much higher O/C range, with abundant average $OS_C$ in the range of -1 to 0, as well as abundant low O/C
and low H/C compounds.
The number of unique nitrogen-containing compounds is almost identical for both sites, but the relative nitrogen
number distribution is shifted to lower (or zero) nitrogen numbers for PM of Bely Island. The opposite trend is
observed for the oxygen number distribution. The unique peaks detected in PM from Bely Island show a much
higher degree of oxygenation (maximum at 13 oxygen) whereas most of the unique peaks in Nadym city PM
samples contain less than 6 oxygen atoms. The overlap of both datasets is observed in between both oxygen number
distributions.
These characteristic differences between PM in Nadym city and on Bely Island wildfire plume agree with the
previously discussed more intensively atmospheric aging of the organic aerosol in wildfire PM collected at Bely
Island, compared to the relatively fresh aerosol collected in Nadym city.

**3.8 Elucidation the origin of individual elemental compositions by Hierarchical Clustering**

Hierarchical cluster analysis is applied to better understand which compounds of the tens of thousands of identified
elemental compositions are the most relevant for the characterization and differentiation of the observed sample
origins (wildfire plumes sampled in Nadym city and Bely Island, mixing of plume with gas flaring emissions, and
samples after the wildfire plume had passed). Considering all elemental compositions, including both, [M+H] and
[M+Na] adducts, that are found in ESI+ datasets, HCA (with max. number of clusters set to 5) is performed to sort
each elemental composition into a separate cluster (average silhouette value = 0.40) (Merder et al., 2021). The
resulting clustergram (Fig. S12) shows the grouping of samples with known similar origin into the same clusters.
In order to highlight molecules related to different sample origins, the elemental composition clustering results are
summarized into five main clusters. The chemical characteristics of the results are visualized in Van Krevelen
space and compound class distribution plots (Fig. 10).
The analysis of elemental compositions by HCA emphasizes the dominating influence of wildfire emissions on
the complex chemical composition of the detected organic aerosol species. Compounds present in the ambient
aerosol samples are also observed as a constant background in the PM samples affected by wildfires at both
sampling sites. The clustering in combination with knowledge of the respective dominating emission sources for
each sample, allows for a deeper discussion of the identified clusters.
Compounds of cluster 4 are detected with high abundance (intensity and number) in the wildfire affected PM at
Bely Island, but are also, less dominating, present in the Nadym PM samples with wildfire influence (Fig. S13).
The VK plot (Fig. 10) shows a broad distribution of compounds with O/C ratios up to unity. The complex pattern
shown by these compounds is in line with the previous finding of intensively aged biomass burning aerosol arriving
at Bely Island from 31 July to 06 August 2021. Cluster 2 contains typical compounds present in wildfire affected




PM sampled in Nadym city, which are rarely present in PM on Bely Island with and without wildfire influence.
The pattern shown by compounds in this cluster is less distributed over the VK-space, with lower O/C ratios and
higher H/C ratios, compared to the SOA compounds in cluster 4. This suggests less intense atmospheric aging of
the wildfire plume arriving to Nadym from 05 to 08 August 2021, thus PM in Nadym city still had a substantial
content and clear signature of fresh biomass burning emissions. In addition to the fresh biomass burning emissions,
aged aerosol species (cluster 4) are also detected. The detection of both aged and fresh biomass burning emissions
agrees with the observation of higher aerosol absorption in the part of the wildfire plume transported to Nadym
city than transported to Bely Island. A high wildfire plume density suppresses photochemistry inside of the plume
due to light absorption, lower OH radical production and lower ratio of atmospheric oxidants to reactive aerosol
species. Therefore, cluster 2 and 4 contain similar numbers of sum formula in PM samples affected by wildfire
aerosol in Nadym city, but substantially more sum formulas in cluster 4 in PM samples affected by wildfires in
Bely Island.
The impact of gas flaring on sample N07 is clearly represented by compounds in cluster 1 (red), as it is the only
sample where cluster 1 is significantly populated. Compounds of this cluster are found in the highly aromatic, low
O/C ratio region of the VK-space, as was previously discussed for the gas flaring impacted sample. Compounds
from typical concentration levels in Nadym city and on Bely Island, termed as ambient aerosol samples (N08–N10
and B03) are grouped into clusters 3 and 5, respectively. Cluster 5 has its highest contribution to the Bely sample
without wildfire impact, while compounds from cluster 3 are found in most PM samples of Nadym city with
highest relative contribution after the wildfire plume had passed the site.
**4 Conclusions**
Our study shows the long- range transportation of wildfire plume over different trajectories and provide insights
into the chemical composition of aged air pollutants in the Siberian Arctic. Due to PM sampling in Nadym city
and at Bely Island in north of Western Siberia at the same time, it was possible to observe the different atmospheric
fate of the plume periphery and center aerosol at similar atmospheric residence time and transport distance. First,
back trajectory analysis with in-depth chemical characterization of the organic compounds by complementary
mass spectrometric techniques revealed a complex organic mixture of primary and secondary organic aerosols at
both sites, and confirmed the dominant biomass burning source for the samples N01 to N07. In situ detection of
resinoic acids and alteration products as pyrolytic retene in relation to primary retene specified the biomass to
coniferous vegetation and possibly provides additional indication of the combustion efficiency in biomass burning.
Furthermore, the additional influence of regional gas flaring on sample N07 could be underlined from its
contribution of larger PAHs to the PM burden, which may serve as a criterion to separate contributions to the
Arctic PM burden by wildfire from anthropogenic combustion emissions. After the plume passed Nadym, neither
the molecular signature of BB nor gas flaring was found in the samples N08 to N10 with up to one order of
magnitude lower concentrations of OC and EC compared to samples N01 to N07.
During the main plume period at both sites, CHO and CHON were the dominating compound class observed in
ultra-high resolution mass spectra by all ionization techniques, with especially nitrogen-containing compounds
being of interest due to their effect on the light absorbing properties of aerosols. Nevertheless, both sites showed
distinct differences in their more detailed chemical properties. PM samples from Bely Island were more oxidized



with a higher oxidation state, but lower aromaticity than PM samples from Nadym city. The biomass burning
aerosol arriving at Bely Island was identified to originate from the plume periphery of lower concentration, thus it
underwent more intense atmospheric aging than the center plume transported to Nadym city, despite similar
physical plume ages. Finally, hierarchical clustering of the ultra-high resolution mass spectra from ESI+ could sort
the detected sum formulae and deconvolute the chemical composition according to contributing aerosol sources.
The long-range transport of a wildfire plume from central Siberia was observed as an unprecedented event of
carbonaceous aerosol influx to the vulnerable Arctic ecosystem. Typical ambient concentrations of OC and EC in
Nadym city and at Bely Island were exceeded by one to two orders of magnitude. Moreover, $AAE_{405/808}$ from 1.5
to 3.3 suggested the presence of BrC in Nadym city, but the weekly average of $AAE_{405/808}$ over a similar period at
Bely Island accounted for 1-1.2, indicating more intense atmospheric aging and degradation of BrC chromophores
from the same wildfire plume.
Despite the known impact of wildfire plumes on the Arctic aerosol composition, the investigated PM samples from
Nadym city and Bely Island describe a long-range transport event with unprecedented high concentrations of
carbonaceous aerosol. Detailed chemical characterization of aged wildfire aerosol emissions provides insights into
biomass burning and atmospheric processes, and may improve our understanding of interactions between the bio-
and atmosphere as well as consequences on the Arctic ecosystem and climate.
**Supplement**
The supplement related to this article is available online at: …
**Author Contributions**
E.S.: Methodology, Investigation, Formal analysis, Writing - Original Draft, Visualization; H.C.:
Conceptualization, Investigation, Formal analysis, Writing original draft, Visualization, Supervision, Project
administration; O.P.: Conceptualization, Investigation, Data curation, Writing original draft, Supervision, Project
administration, Funding acquisition; V.K.: Data curation, Resources, Writing - Review & Editing; M.C.: Software,
Formal analysis, Visualization, Writing - Review & Editing; N.K.: Investigation, Writing - Review & Editing; T.
M.: Investigation, Writing - Review & Editing; C.P.R.: Methodology, Software, Resources, Writing - Review &
Editing, Supervision, Project administration; R.Z.: Resources, Writing - Review & Editing, Funding acquisition
**Competing interests**
The contact author has declared that none of the authors has any competing interests.
**Financial Support**
This work was supported by the German Research Foundation (DFG) under grant ZI 764/24-1. Funding by the
Horizon 2020 program for the EU FT-ICR MS project (European Network of Fourier-Transform Ion-Cyclotron-
Resonance Mass Spectrometry Centers), Grant agreement ID: 731077 is gratefully acknowledged. The authors
thank the DFG for funding of the Bruker FT-ICR MS (INST 264/56). Methodology of air mass transportation and
satellite images analyses is developed under Russian Science Foundation (RSF) № 22-17-00-102. Data collection



and treatment was funded by a grant of the Ministry of Science and Higher Education of Russian Federation under
the Agreement (075-15-2021-574). We acknowledge funding by DFG and University of Rostock for covering the
open access cost via the project 512855535.
**Acknowledgment**
This research was performed according to the Development program of the Interdisciplinary Scientific and
Educational School of Lomonosov Moscow State University 《Future Planet and Global Environmental Change》.

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

**Table 1: Sum parameter trends comparing wildfire plume datasets from Nadym and Bely Island of the same period (31**
**July to 07 August 2021, "<": increase, "≈": steady, ">": decrease).**

| Nadym → Bely | APPI | | | ESI- | | | ESI+ | | |
|---|---|---|---|---|---|---|---|---|---|
| | Nadym city | | Bely Island | Nadym city | | Bely Island | Nadym city | | Bely Island |
| AI < 0.25 [% int.] | 55.60 | < | 66.45 | 69.22 | < | 75.96 | 86.09 | < | 88.56 |
| AI > 0.5 [% int.] | 6.83 | > | 4.48 | 2.71 | < | 3.40 | 0.87 | < | 2.12 |
| DBE | 8.07 | > | 7.09 | 7.70 | > | 6.75 | 5.28 | < | 6.01 |
| $OS_c$ | -0.80 | < | -0.61 | -0.50 | < | -0.32 | -1.09 | < | -0.78 |
| log(C*) | -5.41 | < | -5.19 | -8.01 | < | -4.99 | -5.49 | ≈ | -5.87 |
| O/C | 0.31 | < | 0.42 | 0.48 | < | 0.54 | 0.30 | < | 0.40 |
| O/N | 5.78 | < | 6.65 | 6.66 | < | 8.39 | 4.70 | < | 6.83 |
| CHO [% int.] | 47.12 | < | 48.10 | 44.05 | < | 64.70 | 45.96 | < | 51.40 |
| CHNO [% int.] | 47.76 | > | 44.10 | 36.71 | > | 28.60 | 47.38 | > | 45.30 |

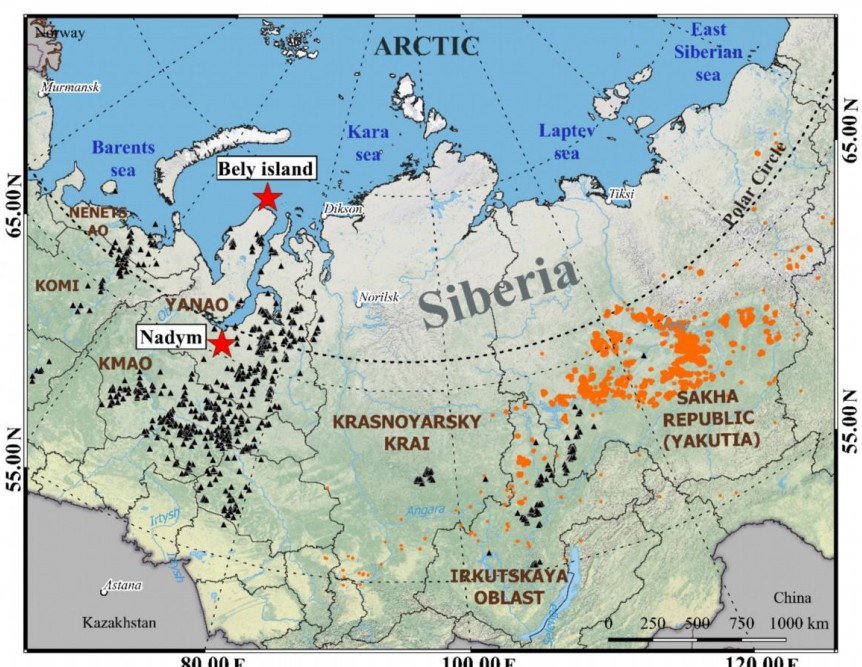

**Fig.1. Location of Bely Island (73°20′7.57″N, 70°4′49.05″E) and Nadym city (65°32′00″ N, 72°31′00″ E) in Yamalo-**
**Nenets Autonomous Okrug (YNAO) (western Siberia). The map was created using Open Source Geographic**



Information System QGis (https://qgis.org/en/site) with ESRI Physical imagery
(https://server.arcgisonline.com/ArcGIS/rest/services/World_Physical_Map/MapServer/tile/%7Bz%7D/%7By%7D/
%7Bx%7D&zmax=20&zmin=0) as the base layer. Moreover, open-source Natural_Earth_quick_start package was
used to add a layer of natural and cultural boundaries and polygons from ESRI Shapefile storage. VIIRS and MODIS
active fire data for August 2021 are downloaded from https://firms.modaps.eosdis.nasa.gov/, shown by orange. Flares
of oil and gas fields are indicated for 2020 as black triangles (https://skytruth.org/, last update 2023).

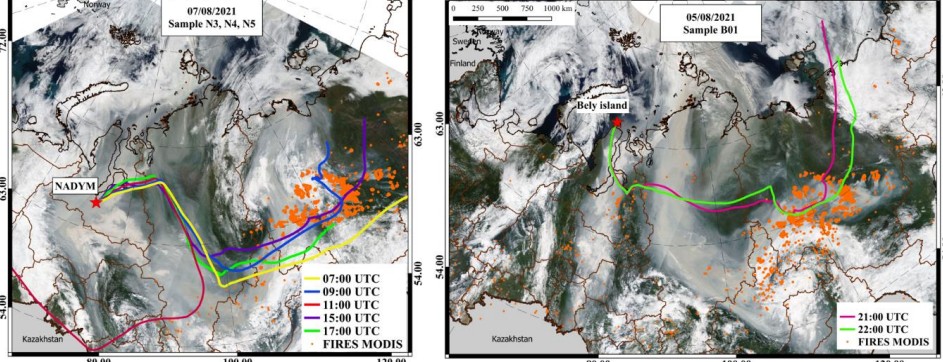

**Fig.2. Backward trajectories and satellite image of smoke plume on the day of peaking concentrations of carbonaceous**
**aerosols for Nadym city on 07 August 2021 (left) and Bely Island on 05 August 2021 (right). Dates of sample completion**
**for N03, N04, and N05 from Nadym city and B01 from Bely Island are presented. The maps were created using Open**
**Source Geographic Information System QGis (https://qgis.org/en/site) with satellite imagery from 07 and 08 of August**
**2021 (https://worldview.earthdata.nasa.gov) with TERRA MODIS fire anomaly layer and The MODIS Corrected**
**Reflectance true color imagery as the base layer. (MODIS Science Team, 2017d, 2017c, 2017b, 2017a) Open-source**
**Natural_Earth_quick_start package was used to add a layer of natural and cultural boundaries and polygons from**
**ESRI Shapefile storage. Backward trajectories were calculated using HYSPLIT software**
**(https://www.ready.noaa.gov/HYSPLIT.php).**

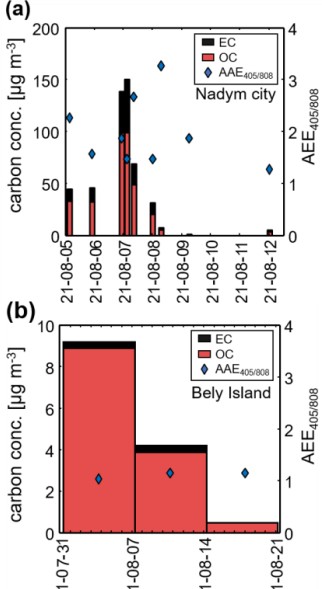



**Fig. 3. Carbon concentrations and Angström absorption exponent (AAE₄₀₅/₈₀₈ₙₘ) in Nadym (a) and on Bely Island (b)**
**measured by a multi-wavelength thermo-optical carbon analyzer (TOCA). Sampling conditions as well as EC and OC**
**values, divided into individual fractions based on the Improve_A protocol, are listed in Table S1.**

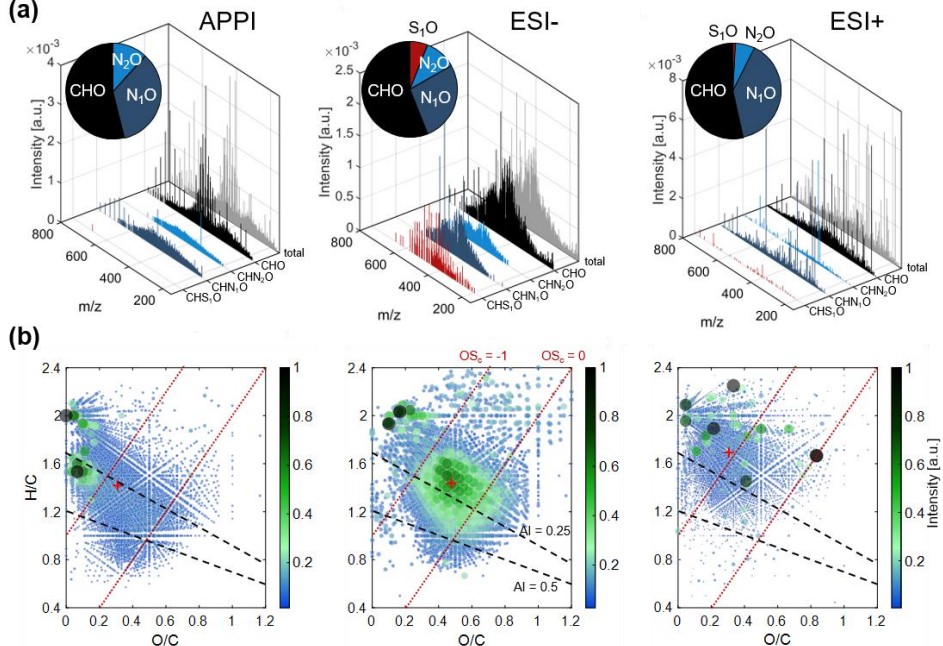

**Fig. 4: FT-ICR MS data overview. A: Averaged and normalized assigned elemental composition mass spectra of wildfire**
**plume impacted samples in Nadym (N01–N05) in three ionization techniques (left to right: APPI, ESI- and ESI+), with**
**insert of relative abundance pie chart of four main compound classes. B: Van-Krevelen diagrams with average intensity**
**weighted H/C and O/C ratio marked red. Dotted lines indicate limits of average carbon oxidation state (OS_C, red) and**
**modified aromaticity index (AI, black).**

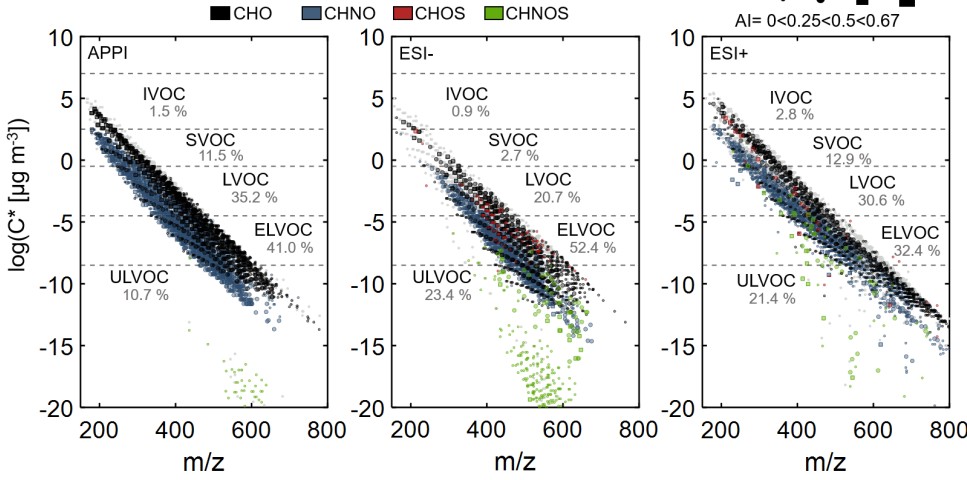

**Fig. 5: Saturation vapor pressure (C\*) versus m/z plot for unique compounds from averaged wildfire impacted samples**
**N01–N05, separate by compound class (black: CHO, blue: CHNO, red: CHOS, green: CHNOS). Compounds abundant**
**after strong wildfire impact (sample N08-N10) in grey. Aromaticity index indicated by dot shape and size. Relative**
**number of compounds per volatility class listed below each volatility class label.**



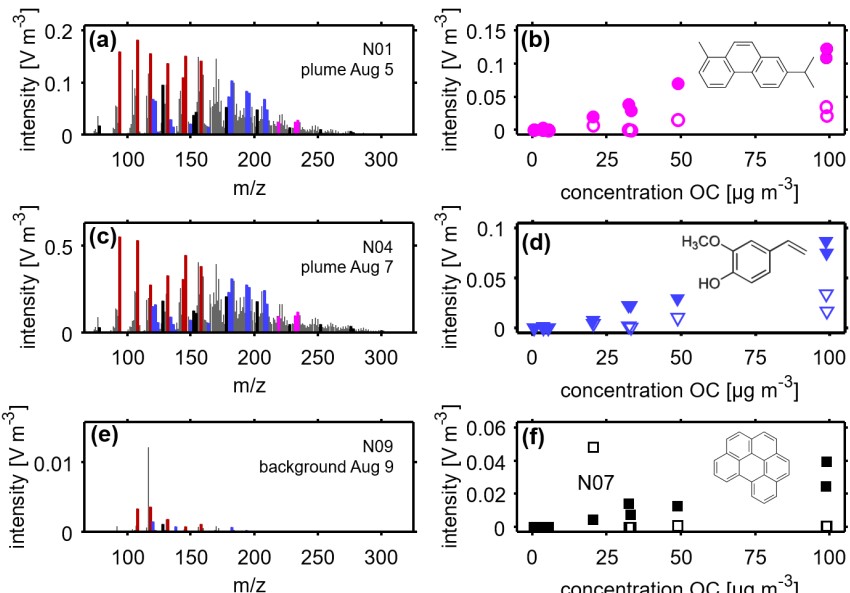

**Fig. 6: REMPI mass spectra, normalized to sampling volume, of OC3-4 from Nadym city samples N01 (A) and N04 (C)
with elevated OC and EC concentrations as well as N09 as reference for PM without wildfire impact (E). Different
colors indicate decomposition products typical for coniferous plants (magenta), cellulose/SOA (red) and lignin (blue) as
well as parent PAH (black). Correlations of (B) m/z 234 (e.g. retene), (D) m/z 150 (e.g. vinylguaiacol) and (F) m/z 276
(e.g. benzo[g,h,i]perylene) in OC1-2 (open symbols) and OC3-4 (filled symbols) to OC.**

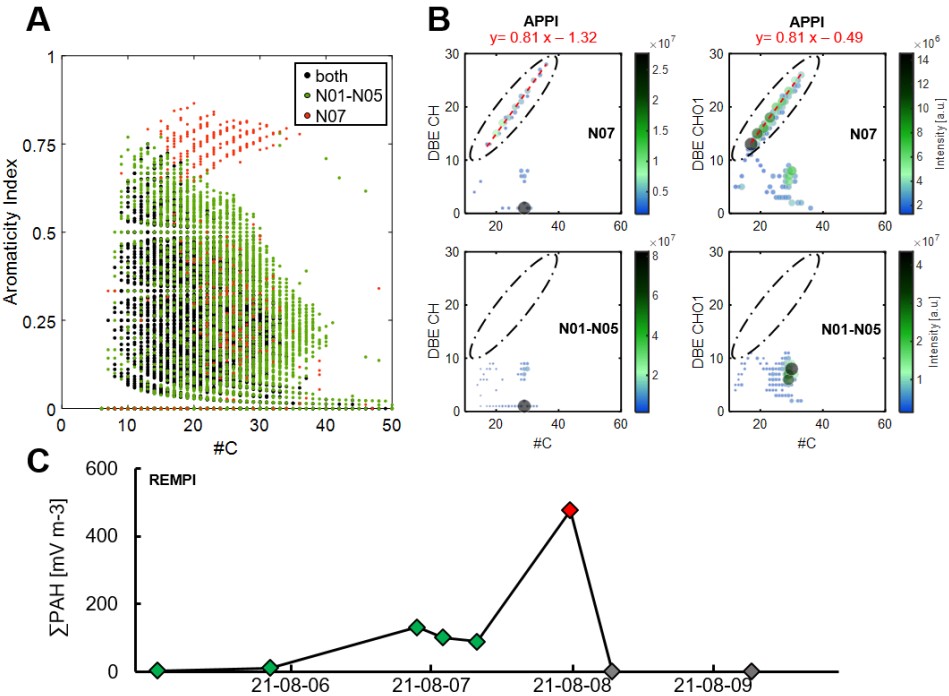

**Fig. 7: A: Aromaticity index (AI) versus carbon number plot of sum formulae (APPI) observed uniquely in the wildfire**
**impacted samples at Nadym (N01–N05, green), in sample N07 additionally impacted by gas flaring (red) or both datasets**
**(black). B: Double bond equivalent (DBE) versus carbon number plot of CH (left) and CHO₁ (right) compound classes**
**for main plume (bottom) and N07 (top) with linear equation of planar limit indicated in red. C: Time trend of summed**
**intensities of parent PAHs related *m/z* (128, 152, 178, 202, 228, 252, 276, 278, 300 and 302) detected by TOCA-REMPI-**
**TOFMS in OC1-2.**

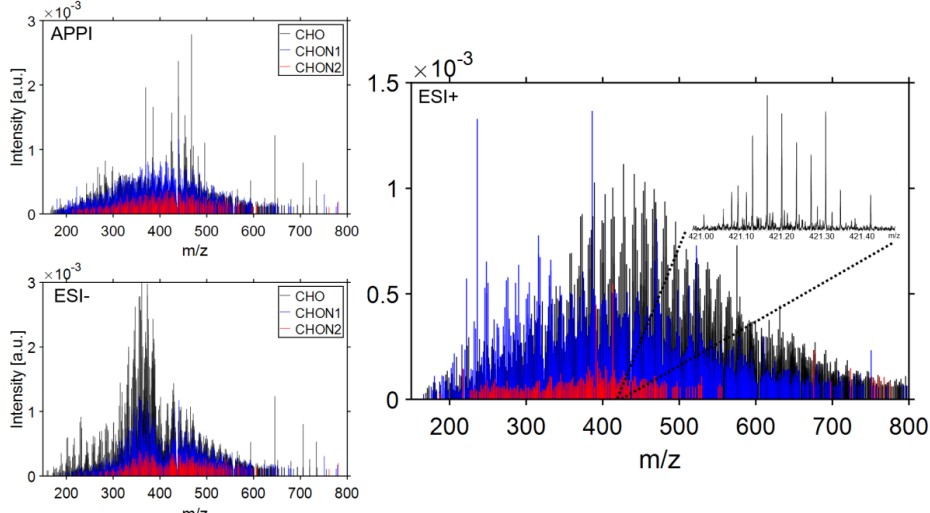

**Fig. 8: TIC normalized mass spectra of assigned CHO (black), CHN1O (blue) and CHN2O (red) elemental compositions**
**identified in the wildfire affected PM sampled at Bely Island, collected during the same period as wildfire aerosol was**
**arriving to Nadym city, by APPI and ESI. Zoom-in on one nominal mass of the ESI+ spectrum highlighting the**
**molecular complexity of oxygenated compounds at m/z 421.**



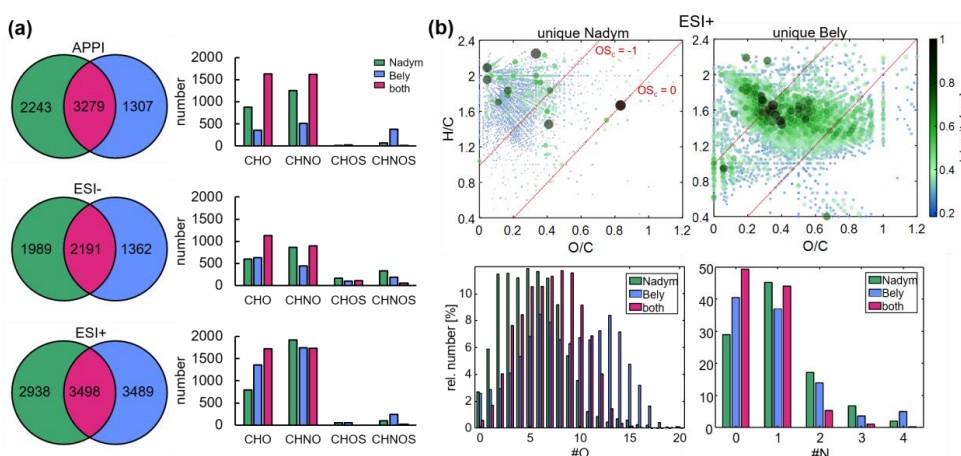

**Fig. 9: A: Venn diagram with sum formula numbers (left) and relative intensity compound class distribution (right) of Nadym main plume (green) and Bely Island (blue) wildfire aerosol datasets. B: Van-Krevelen plot of unique compounds (ESI+) identified in Nadym city and Bely Island PM samples and relative number distribution of oxygen- and nitrogen-containing compounds for unique and shared (magenta) elemental compositions.**

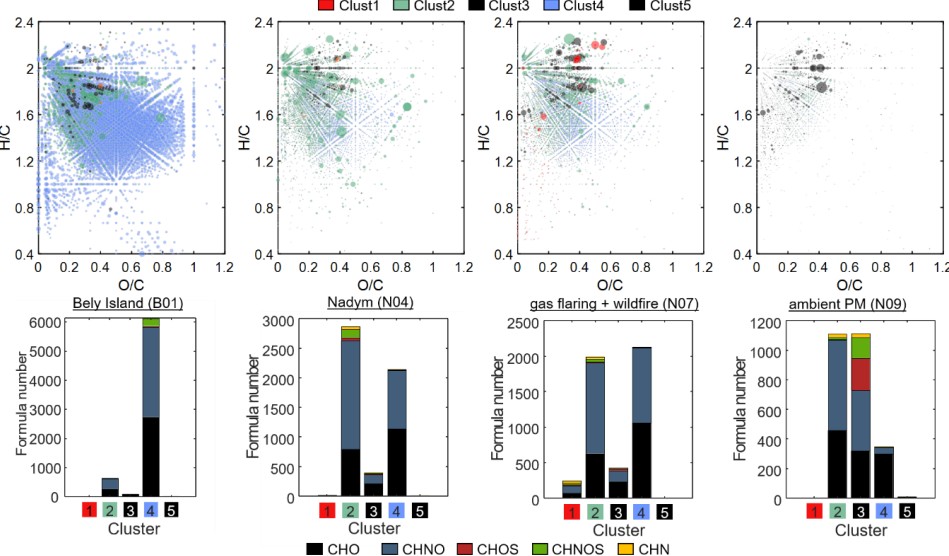

**Fig. 10: Visualization of hierarchical cluster analysis (HCA) results based on all elemental compositions detected in ESI+, with the number of clusters set to five. Van-Krevelen (VK) plots of four exemplary samples and bar plots of the compound class distribution in each cluster. The colors in the VK plot indicate the cluster assignment of each elemental composition, with dot size indicating intensity of each compound. Cluster 4 (blue) is associated with aged wildfire compounds, cluster 2 (green) is associated with more fresh wildfire aerosol, cluster 1 (red) with gas flaring and cluster 3 and cluster 5 (black) are associated with ambient PM without contribution of wildfires at Bely Island and in Nadym, respectively.**

