# Peer review of "Comprehensive mass spectrometric analysis of unprecedented high levels of carbonaceous aerosol particles long-range transported from wildfires in the Siberian Arctic"

_EGUsphere, 2023_

## Author Comment (AC1)

**Reviewer 1**

This manuscript presents mass spectrometric analyses of carbonaceous aerosol in long-range transported smoke from Siberian wildfires. The analysis is based on off-line analysis of filter samples collected at two sites in Northwest Siberia: Nadym city and Bely island, which lies 800 km further north. In August 2021, both sites were impacted by smoke transported from wildfires covering large areas in Yakutia, approx. 2000 km east.

Chemical analysis of the organic aerosol is comprehensive, with three different mass spectrometric methods applied to the samples. Furthermore, multiwavelength light absorption is included in the analysis. On the other hand, there are no supporting measurements such as trace gases (e.g. CO NOx) to augment the analysis. Also, the Hysplit back-trajectories accumulate rather large uncertainty for such long distance as covered here.

Overall, this manuscript offers a relatively rare picture of carbonaceous aerosol from Siberian wildfire emissions and I recommend it for publication after some minor comments have been addressed.

Response: We thank the reviewer for undertaking this review and support for publication. In following, we try to address the reviewer's comments.

Minor comments

I do not think it is correct to refer to the smoke from the remote fires as one plume. Especially, interpreting the differences between the two sites through dilution / plume edge effects only seems oversimplification. Given that the sites are 800 km apart and wildfires were occurring over an area of several hundred km across it is quite possible that there were differences in the emissions as well as in the processing during the transport.

Response: We basically agree with the reviewer's view on plume's spatial occurrence and that the concept of plume edges and center is a simplification. It does not necessarily relate to a single plume, rather than it describes the spatial smoke density with a "plume center (or centers) at substantially elevated aerosol concentration and edges approaching more typical ambient aerosol levels without clear definition. The benefit of this classification is that it accounts for the evaporation-oxidation mechanism for secondary aerosol formation, (Palm et al., 2020) which is increased in relevance with increasing aerosol concentration. Nevertheless, we would like to point out that in fact the OMPS Aerosol Index shown in Fig. S2 demonstrates a rather large cohesive area of constantly high aerosol concentration instead of multiple segregated individual plumes at the given spatial resolution of 50km x 50km.

Referring to Fig. 2 of the manuscript, air masses moving both to Nadym and Bely crossed the fire site from East to West, thus they likely took up similar emissions. Additionally, the forest area of Central Yakutia has similar vegetation, largely dominated by larch and pine trees.. (Kharuk et al., 2021) Hence, we can assume that despite widespread wildfires, there was an uptake of rather similar BB aerosol over the intersection of the relevant air masses and the BB area, justifying the consideration of the emission as single large plume.

Based on Fig. 1, there is extensive gas and oil field flaring between Nadym and the wildfire area, but less so for Bely island. Here, more detailed air mass history analysis with higher resolution meteorological data (e.g. Flexpart with ERA5) would have helped to show if smoke observed at both sites had similar exposure to gas flaring emissions. Also, NOx from a trace gas analyser, had it been available, would have been useful. For instance, Table 1 shows that Nadym had higher CHNO fraction than Bely island. Is this due to higher NOx from the flaring, or due to different chemistry in a more concentrated plume?

Response: We can expect that gas flaring contribute to PM in both Nadym and Bely, but quantitatively depending on the meteorological conditions. Based on the BWT analysis in Fig. 2, also the air masses traveling to Bely reach first the gas fields of YNAO before moving North to Bely.

Unfortunately, the sampling was initiated because of public news about larger fires in Yakutia and arriving smoke in Nadym, and not as part of a measurement station for air quality monitoring, so only PM samples are available, but no gases could be measured.

We cannot ultimately confirm or rule out if different gas flaring contributions of NOx or the impact of different concentrations inside the plumes of Bely and Nadym are responsible for the different CHNO relative contributions. However, relative difference in CHNO between Nadym and Bely were less than 4 percentage points for APPI and ESI+, and about 8 percentage points for ESI-. APPI and ESI with direct infusion and FT-ICR MS is not an analytical approach for quantitative analysis and based by matrix effects. The reason for the number of 4 significant digits given is the fact that the measurements have the precision when repeated. Furthermore, the comparison in Table 1 does not cover exactly the same time periods, so results of minor difference cannot be discussed.

Specific comments

P1, L37-38 "Owing to lower aerosol concentration in the plume periphery than in its center, it is demonstrated how dilution affects the chemical plume composition during atmospheric aging." The sites are 800 km apart. As commented above, changes in the emission or effect of additional NOx from gas flaring have not been ruled out.

Response: We thank the reviewer for pointing this out. The BWT analysis shows that the air mass to Bely Island and Nadym city proceeded in parallel with few degrees difference in latitude until the main gas flaring area was reached. Afterward, parallel BWTs divided and went northern to Bely Island and western to Nadym city. Therefore, we are convinced that both air masses received similar mixing with gas flaring aerosol.

P3, L20-21 "the phenomenon of dry thunderstorms, which have been estimated to account for more than a half of the fire causes." Please provide reference.

Response: A reference (Narita et al (2021)) for this statement is given in the sentence following.

P3, L30-31 "differences in atmospheric aging of plume center and plume periphery." Please see previous comments on other possible causes for the differences between the sites.

Response: We agree that the atmospheric fate of the plume is a sum of different conditions, including interference with other sources. However, the concentration of particulate matter is one to two

orders of magnitude higher than in reference samples (i.e., when the plume had passed). Additionally, processing of the plume did not only take place in the gas production area, but already in a distance of >2000km in a remote area. Along the trajectories of the wildfire plume to Nadym and Bely, between approximately 60 and 67° latitude, there is no larger city in the Republic Sacha and Krasnoyarsk Krai, or known larger emission sources from industrial production. Gas flaring aerosol may be contributed not before crossing the border from Krasnoyarsk Krai to YNAO. Therefore, we conclude that the majority of the atmospheric residence of the plume happened under conditions with low or even no influence of gas flaring. After entrance to YNAO, both plumes (to Bely and Nadym) were first mixed with gas flaring emissions, but were subsequently processed for approximately 1.5 days (see exemplary figure for samples N03/N4 and B01 below) before arriving at the respective sampling sites.

[Figure]

[Figure]

P4, L36 Please explain in more detail what the different OC and EC components (Table S1) from IPMROVE_A protocol represent and what are the assumptions and uncertainties related to determination of e.g. OCpyro.

Response: We omitted further details on the carbon analysis because it is a well-established method in air quality monitoring programs and emission. (Chow et al., 2007) Despite the multiwavelength carbon analyzer is a rather new instrument, the uncertainties of the carbon fractions remain unaffected compared to the single-wavelength instrument. (Chow et al., 2015) Generally, uncertainties are sample-dependent, typical values are 2-6% for OC and TC, and 5-10% for EC; largest uncertainties are usually found in the volatile fraction OC1 and $OC_{pyro}$ of about 50. (Zhang et al., 2021) Regarding uncertainties to the optical measurement, we refer to Chow (Chow et al., 2021) and provide an overall uncertainty of the presented Angström Absorption Exponent (AAE), calculated from the precision of the laser transmittance measurement described in. (Chen et al., 2015)

Generally, the numbering of the OC and EC fractions refers to increasing thermal refractiveness of the carbon, for OC corresponding to the reciprocal volatility. As a rough classification, in OC1 and OC2, particle-bound species are mainly thermally desorbed, while in OC3 and OC4, thermal decomposition prevails the volatilization.

We added information to the uncertainty of carbon quantities p. 4 l. 37:

*"... for the separation of pyrolytic OC from EC. Precisions of the carbon analysis is sample-dependent and range between 2 and 6% for TC and 5 to 10% of the split between OC and EC according to the manufacturer. In addition to 635 nm,..."*

P6, L4 "HYSPYT" do you mean HYSPLIT?

Response: Yes, the spelling is corrected to "HYSPLIT".

P6, L22-24 "The OMPS Aerosol Index (Fig. S2) suggests that the periphery (lower OMPS Aerosol Index, yellow) of the Yakutian wildfire plume was transported to Bely Island in contrast to the plume center aerosol transported to Nadym (higher OMPS Aerosol Index, red)." Please give numerical values for the aerosol index. Please also change Fig. S2 so that the colorbar is legible.

Response: We agree with the reviewer and have added the numerical values of the OMPS (e.g., 21-08-07: Bely 1.85, Nadym >5) to the text as well as adapted the size of the colorbar in Fig. S2 to improve visibility.

P6, L24-27 "This may have led to a gradient in photochemical processing of the plume, i.e., a lower extent of atmospheric processing by OH radicals, with the northern section containing more atmospherically aged aerosol and the southern section more fresh wildfire emissions, which were picked up on the way westward." Previously, the effect of decreased OH concentration has been shown for fresh plumes (age some hours) only. In this case, is the difference in actinic flux large enough to have an effect after maybe a week or more of atmospheric ageing?

Response: We thank the reviewer for this question. First, we based our statement on the fact that trajectories of the air masses to Nadym and Bely Island were largely parallel with some higher degrees in latitude for the latter one as shown in Fig. 2. OMPS data show evidence for substantially lower aerosol concentration in the air masses to Bely Island than to Nadym, so the apparent difference is a significant dilution factor between the two parts of the plume, supported by the OC concentrations in Nadym and Bely Island with a factor of 10 difference. Despite anticipated dilution over the entire transported distance, the OA arriving at Nadym still has a concentration > 100 $\mu g/m^{-3}$. In Hodshire et al. (2021) aging of several wildfire plumes is shown starting from less than 1 hour of aging with a corresponding OA concentration range from 10 to 650 $\mu g/m^{-3}$ (Hodshire et al., 2021). Therefore, it is likely that aging of the plume involved differences in edge/periphery and core/center of the plume over long distances of transportation. Furthermore, we are convinced that the effect of different OH production rates caused by transportation at different latitudes and consequently different actinic fluxes is of minor importance and not exceeding one order of magnitude difference in aerosol concentration and related OH reactivity.

We agree with the reviewer that suppression of OH appears in fresh plumes. The referenced sentence describes the uptake of wildfire aerosol in Yakutia by the modeled air mass, so indeed it refers to rather fresh and near-source BB aerosol. However, because of the large dimension of the fires, sources are considered to cover a distance of up to 500km.

BB aerosol aging proceeds rapidly and significant changes in aerosol composition and concentration are visible within in the first few hours after emissions. Our analytical techniques are consistently showing that the PM in Nadym still contains a molecular signature of BB, whereas in the PM of Bely Island BB are virtually absent. Considering the similar source and transportation of Yakutian wildfire PM, it points towards the fact that plume aging conditions may apply over longer aging times than hours, so the concepts in aging of fresh plumes are still applicable for longer physical ages.

P7, L12-15 "Regarding biomass burning, spectral absorption obtained throughout the near-ultraviolet to near-infrared spectral region and high Angstrom absorption exponents (AAE) up to 4.4 are were found for

smoke from smoldering combustion of pine debris in the wavelength regions from 370 to 670 nm." Please provide reference.

Response: The reference (Popovicheva and Kozlov, 2020) in the following sentence applies to both flaming and smoldering aerosol optical properties.

P7, L37 Elevated NOx may increase BrC formation during atmospheric ageing. Thus, if there are differences in the gas flaring or other NOx mixing into the plume, that could explain part of the difference. Thus, it is not self-evident that the difference in AAE is due to photobleaching only.

Response: The paragraph from L28 to L37 on page 7 deals with the AAE of the PM collected at Bely Island, where weekly averages of 1.0 and 1.2 were obtained, which is a typical result for this location. (Popovicheva et al., 2022) AAE of the high OC and EC concentrations in Nadym appeared from 1.5 to 2.7, which is a reasonable range for wildfire plumes despite different measurement techniques and wavelengths used. (Selimovic et al., 2020; May et al., 2023) The sample of supposed gas flaring contribution (N07) revealed an AAE of only 1.5.

We agree with the reviewer that significant NOx contributions may increase BrC concentration by formation for nitrogen-containing compounds, like nitrophenols, which are known as strong chromophores. As indicated by our BWT analysis, both air masses going to Bely Island and Nadym city passed through the gas flaring fields, thus were mixed with similar gas flaring aerosol emissions incl. $NO_x$. A significantly more intense mixing of BB aerosol going to Nadym city with gas flaring emissions is not evident.

AAEs measured for Bely Island are lower than the AAE of sample N07 and any other plume-affected sample from Nadym. As evident from several BB aerosol ageing studies, BrC was released as primary emissions or was formed after short ageing e.g., (Laskin et al., 2015), but to reach the AAE at Bely Island, degradation by photobleaching must have been taken place.

We noticed a higher contribution of most likely gas flaring in sample N07 based on the signature of polycyclic aromatic hydrocarbons (PAH). However, apart from larger PAH, the chemical composition of N07 with plume samples arriving at Nadym city before (N01-N05) showed a high similarity, supporting that different mixing with gas flaring emission had a negligible effect on the composition of the BB plume.

P8, L25 "time period (21-07-31 to 07-08-21)" Please check date.

Response: We have corrected the date to 21-08-07.

P9, L4-5 "The majority of compounds is found in the low to ultra-low volatility area, but there is a difference when comparing individual compounds classes." Please discuss the uncertainty in the volatility parameterisation.

Response: As the calculation of the saturation vapor pressure (log(C*)) in this study is based on the elemental composition, the uncertainty for the individual saturation vapor pressure value may be moderate, as it is based on fitting by multi-linear least squares analysis of 30,000 model compounds. (Li et al. ACP 2016) However, the parametrization and separation into volatility classes (e.g., intermediate-volatile C*=3x10$^{-6}$ - 300 µg m$^{-3}$, semi-volatile C*=300 - 0.3 µg m$^{-3}$) is based on volatility classes that span a several orders of magnitude. Therefore, the volatility parametrization can be considered as sufficiently accurate for these purposes.

P9, L35-36 "This has been assigned to HU-HOM, which are produced from the photooxidation of larger PAHs on soot particles, thus indicating heterogeneous processing of wildfire aerosol particles." Please provide reference.

Response: The reference (Li et al, 2022) in the following sentence applies to definition ad origin of HU-HOMs.

P12, L36 – P13, L40 Please consider splitting the long paragraph into shorter ones.

Response: We agree with the Reviewer and split the paragraph.

P13, L16-21 "According to RETprim/RETtot close to zero, samples N01–N02 (06 August 2021) contain biomass burning aerosol e.g., originating from fires at smoldering condition (Fig. S6). From 07 August 2021 to the morning of 08 August 2021 (N03–N07), the fires became more intense and turned over to more flaming conditions, suggested by increased OC and EC concentrations, lower ratios OC-to-EC being typical for higher combustion efficiency, and RETprim/RETtot between 0.15 and 0.26; on these days, the main plume by means of highest aerosol concentrations arrived Nadym city." Did you observe any difference in the mass spectra that could be explained by the apparent differences in the combustion characteristics? For instance, Sekimoto et al. (2018, 2023) found distinct VOC profiles for the high and low temperature combustion.

Response: We thank the reviewer for this question. We are aware of the concept outlined by Sekimoto et al. and developed a similar model for wood stove emissions based on a 3-factor solution. (Czech et al., 2016; Elsasser et al., 2013) However, all of these studies were based on online measurements, i.e., contained a large number of samples, whereas our dataset is typical for "large p, small n" and thus not suitable for deconvolution.

We may use quantitative ECOC data and AAE of this study in combination with parametrization to the modified combustion efficiency (MCE) by Pokhrel et al (2016) from laboratory burns for deriving combustion conditions. (Pokhrel et al., 2016) Based on the ratio of EC to OC in the plume at Nadym city (samples N01 to N05), we obtain AAEs for the wavelengths 405, 532 and 660nm from 2.1 to 2.3, which agrees well with our AAE data (from wavelengths 405 and 808nm) from 1.5 to 3.0. With an AAE of 2.2, we obtain a MCE between 0.9 and 0.96, indicating rather flaming burning conditions, which agrees with the average MCE for boreal forest fires from Akagi et al. (Akagi et al., 2011)

Overall, the parametrization from Pokhrel et al. generates plausible from our measurements for MCE, but further details about combustion conditions would be too speculative considering the available information. (Pokhrel et al., 2016)

P15, L29-30 "Also, AAE values are decreased, indicating degradation of chromophores by photobleaching (Liu et al., 2021)." Or then secondary BrC was not formed, due to e.g. lower NOx.

Response: We do not state that secondary BrC was formed or not formed. Generally, it is well-known that primary biomass burning aerosol contains BrC species, especially from relatively poor combustion conditions, associated with increased AAE. (Laskin et al., 2015) Regardless if secondary BrC have been formed during atmospheric transport, BrC from the primary BB emissions must have been – likely photochemically considering the meteorology - degraded to reach AAEs close to unity. For longer photochemical ages, photobleaching is a well-known phenomenon that would explain our observations. For PM collected at Nadym city, relatively high AAEs were obtained. Although these AAEs might be caused

by secondary BrC formation, it fits to the parametrization by Pokhrel et al. (2016) for primary BB aerosol and our result of a distinct BB molecular signature. Therefore, we assume that the high AAEs are mainly caused by primary BrC species from the forest fire.

P18, L8-11 "Moreover, AAE405/808 from 1.5 to 3.3 suggested the presence of BrC in Nadym city, but the weekly average of AAE405/808 over a similar period at Bely Island accounted for 1-1.2, indicating more intense atmospheric aging and degradation of BrC chromophores from the same wildfire plume." Please see previous comment.

Response: We would like to point out again that at the source of the BB aerosol, we may assume AAE significantly larger than unity as typical for BB emissions. Photobleaching after long-range transport adequately explains the low AAE measured in the PM of Bely Island while differences in aging between the two sites can be explained by different aerosol concentrations, i.e., different OH reactivities.

**References**

Akagi, S. K.; Yokelson, R. J.; Wiedinmyer, C.; Alvarado, M. J.; Reid, J. S.; Karl, T.; Crounse, J. D.; Wennberg, P. O. Emission factors for open and domestic biomass burning for use in atmospheric models. Atmos. Chem. Phys. 2011, 11 (9), 4039–4072. DOI: 10.5194/acp-11-4039-2011.

Chen, L.-W. A.; Chow, J. C.; Wang, X. L.; Robles, J. A.; Sumlin, B. J.; Lowenthal, D. H.; Zimmermann, R.; Watson, J. G. Multi-wavelength optical measurement to enhance thermal/optical analysis for carbonaceous aerosol. Atmos. Meas. Techn. 2015, 8 (1), 451–461. DOI: 10.5194/amt-8-451-2015.

Chow, J. C.; Wang, X.; Sumlin, B. J.; Gronstal, S. B.; Chen, L.-W. A.; Trimble, D. L.; Kohl, S.; Mayorga, S. R.; Riggio, G.; Hurbain, P. R.; Johnson, M.; Zimmermann, R.; Watson, J. G. Optical Calibration and Equivalence of a Multiwavelength Thermal/Optical Carbon Analyzer. Aerosol Air Qual. Res. 2015, 15 (4), 1145–1159. DOI: 10.4209/aaqr.2015.02.0106.

Chow, J. C.; Watson, J. G.; Chen, L.-W. A.; Chang, M. O.; Robinson, N. F.; Trimble, D.; Kohl, S. The IMPROVE_A Temperature Protocol for Thermal/Optical Carbon Analysis: Maintaining Consistency with a Long-Term Database. J. Air Waste Manag. Assoc. 2007, 57 (9), 1014–1023. DOI: 10.3155/1047-3289.57.9.1014.

Czech, H.; Sippula, O.; Kortelainen, M.; Tissari, J.; Radischat, C.; Passig, J.; Streibel, T.; Jokiniemi, J.; Zimmermann, R. On-line analysis of organic emissions from residential wood combustion with single-photon ionisation time-of-flight mass spectrometry (SPI-TOFMS). Fuel 2016, 177, 334–342. DOI: 10.1016/j.fuel.2016.03.036.

Elsasser, M.; Busch, C.; Orasche, J.; Schön, C.; Hartmann, H.; Schnelle-Kreis, J.; Zimmermann, R. Dynamic Changes of the Aerosol Composition and Concentration during Different Burning Phases of Wood Combustion. Energy Fuels 2013, 27 (8), 4959–4968. DOI: 10.1021/ef400684f.

Hodshire, A. L.; Rammarine, E.; Akherati, A.; Alvarado, M. J.; Farmer, D. K.; Jathar, S. H.; Kreidenweis, S. M.; Lonsdale, C. R.; Onasch, T. B.; Springston, S. R.; Wang, J.; Wang, Y.; Kleinman, L. I.; Sedlacek III, A. J.; Pierce, J. R. Dilution impacts on smoke aging: evidence in Biomass Burning Observation Project (BBOP) data. Atmos. Chem. Phys. 2021, 21, 6839–6855. DOI: 10.5194/acp-21-6839-2021.

Kharuk, V. I.; Ponomarev, E. I.; Ivanova, G. A.; Dvinskaya, M. L.; Coogan, S. C. P.; Flannigan, M. D. Wildfires in the Siberian taiga. Ambio 2021, 50 (11), 1953–1974. DOI: 10.1007/s13280-020-01490-x.

Laskin, A.; Laskin, J.; Nizkorodov, S. A. Chemistry of atmospheric brown carbon. Chem. Rev. 2015, 115 (10), 4335–4382. DOI: 10.1021/cr5006167.

May, N. W.; Bernays, N.; Farley, R.; Zhang, Q.; Jaffe, D. A. Intensive aerosol properties of boreal and regional biomass burning aerosol at Mt. Bachelor Observatory: larger and black carbon (BC)-dominant particles transported from Siberian wildfires. Atmos. Chem. Phys. 2023, 23 (4), 2747–2764. DOI: 10.5194/acp-23-2747-2023.

Narita, D.; Gavrilyeva, T.; Isaev, A. Impacts and management of forest fires in the Republic of Sakha, Russia: A local perspective for a global problem. Polar Sci. 2021, 27, 100573. DOI: 10.1016/j.polar.2020.100573.

Palm, B. B.; Peng, Q.; Fredrickson, C. D.; Lee, B. H.; Garofalo, L. A.; Pothier, M. A.; Kreidenweis, S. M.; Farmer, D. K.; Pokhrel, R. P.; Shen, Y.; Murphy, S. M.; Permar, W.; Hu, L.; Campos, T. L.; Hall, S. R.; Ullmann, K.; Zhang, X.; Flocke, F.; Fischer, E. V.; Thornton, J. A. Quantification of organic aerosol and brown carbon evolution in fresh wildfire plumes. Proc. Natl. Acad. Sci. U.S.A. 2020, 117 (47), 29469–29477. DOI: 10.1073/pnas.2012218117.

Pokhrel, R. P.; Wagner, N. L.; Langridge, J. M.; Lack, D. A.; Jayarathne, T.; Stone, E. A.; Stockwell, C. E.; Yokelson, R. J.; Murphy, S. M. Parameterization of single-scattering albedo (SSA) and absorption Ångström exponent (AAE) with EC / OC for aerosol emissions from biomass burning. Atmos. Chem. Phys. 2016, 16 (15), 9549–9561. DOI: 10.5194/acp-16-9549-2016.

Popovicheva, O.; Kozlov, V. Impact of combustion phase on scattering and spectral absorption of Siberian biomass burning: studies in Large Aerosol Chamber. In Proceedings of SPIE - 26th International Symposium on Atmospheric and Ocean Optics: Atmospheric Physics; Matvienko, G. G., Romanovskii, O. A., Eds. 11560, 2020; p 252. DOI: 10.1117/12.2575583.

Selimovic, V.; Yokelson, R. J.; McMeeking, G. R.; Coefield, S. Aerosol Mass and Optical Properties, Smoke Influence on O 3 , and High NO 3 Production Rates in a Western U.S. City Impacted by Wildfires. J. Geophys. Res. Atmos. 2020, 125 (16). DOI: 10.1029/2020JD032791.

Zhang, X.; Trzepla, K.; White, W.; Raffuse, S.; Hyslop, N. P. Intercomparison of thermal–optical carbon measurements by Sunset and Desert Research Institute (DRI) analyzers using the IMPROVE_A protocol. Atmos. Meas. Tech. 2021, 14 (5), 3217–3231. DOI: 10.5194/amt-14-3217-2021.

---

## Author Comment (AC2)

**Reviewer 2**

Schneider et al. present a detailed assessment of carbonaceous aerosol in Siberia, commenting on topics such molecular-level characterization, plume aging, trajectory analysis, and optical properties. Their multimodal approach used allows for a broad range of molecular coverage, and the findings are well-supported by visuals throughout. While the investigation is well-written overall, I have several concerns detailed below that I believe should be addressed prior to publication.

Comments:

Title: I don't believe that the title of the paper is suitable, as 'comprehensive' and 'unprecedented' are both somewhat misleading. To this point, 'comprehensive' implies a complete characterization of the carbonaceous aerosol. And while the data interpretation herein is indeed detailed, it is largely limited to MS1 observations (i.e., lack of isomeric and structural resolution), and therefore should not be referred to as comprehensive. Use of the word 'unprecedented' is described in a later comment.

> Response: We thank the reviewer for the explanation and removed "comprehensive" from the title. However, the plume transportation event was indeed unprecedented, which we explain more in detail below.

Page 2 Lines 19 – 20: the authors cite a lack of studies on Siberia as motivation and provide a single 2007 reference. However, it appears that there has been more recent work characterizing similar types of samples, and should be cited accordingly (e.g., https://acp.copernicus.org/articles/23/2747/2023/; https://www.sciencedirect.com/science/article/pii/S1352231021000595; https://www.mdpi.com/2072-4292/14/19/4980)

> Response: We thank the reviewer for this comment and try to clarify our intention about the title. Indeed, long-range transport of wildfire aerosol from Siberia does occur and is not an unprecedented event. However, we were unable to find such an event with a) comparable concentrations in respect to the distance of the source and b) apparent transport into the Arctic circle. In May et al. and Johnson et al., Siberian wildfire aerosol was transported over Bering Sea to the US around the polar circle, but did not enter the Arctic. (May et al., 2023; Johnson et al., 2021) We agree that such events appear more frequently. The study by Tomshin & Solovyev (2022) covers the same event as in our study and emphasize the extreme fire season of 2021 associated with high aerosol optical depths and transport toward the North Pole. (Tomshin and Solovyev, 2022)
>
> In the second paragraph of the introduction, we cite two studies on transportation of biomass burning aerosol to the Arctic (Yue 2022; Cali Quaglia 2022) and two about long-range transport of wildfire aerosol specifically from Siberia (Ikeda & Tanimoto 2015; Semoutnikova 2018). The lines 19-20 the referee is referring to put the number of studies from Siberia in relation to other wildfire areas (Flannigan 2009). Furthermore, weather phenomena of Siberia leading to large-scale fires are addressed on page 3 line 17-23 with three more references (Lavoué 2000; Tomshin & Solovyev 2018; Narita 2021). Therefore, we are convinced that we covered wildfires of Siberia to an appropriate extent.

Page 2 Lines 33 – 37: Some more precise langue should be used here.

> Response: We modified the sentence to:
> *"Moreover, heterogeneous reactions between atmospheric oxidants and particle constituents may increase the molecular complexity of primary aerosols in the atmosphere, which is associated with higher functionalization, increase in heteroatom content (O, N, S) and oligomer formation. Additionally, reaction*

*between individual constituents of the particle phase complete the ongoing complex multiphase chemistry (Schneider et al., 2022; Pardo et al., 2022; Lin et al., 2015; Chacon-Madrid and Donahue, 2011)."*

Page 3 Line 5: Instead of using the word 'different', please briefly describe what exactly is different to better contextualize this statement for the reader.

*Response: We changed the statement to "For example, OH radicals, the main oxidant under photochemical conditions, may be already consumed at the periphery of the plume, so enhanced photobleaching was observed at the plume edges relative to the core (Lee et al., 2020), along with faster photochemistry (Palm et al., 2021)."*

Page 3 Line 34: How high off the ground was sampling conducted? Were there any measures taken to minimize potential contribution from dust and/or ground soil particles?

*Response: Filter samples were taken 4 m above sea level, so directly at the ground. Although we cannot rule out the collection of dust and soil particles, their contribution is likely very limited for the PM concentrations illustrated below.*

[Figure]

[Figure]

Smoke enveloping the cities Nadym (left, https://nur24.ru/news/ecologia/smog-ot-pozharov-v-yakutii-polnostyu-okutal-yamal-foto-video) and Noyabrsk (right, https://nur24.ru/news/ecologia/smog-ot-pozharov-v-yakutii-polnostyu-okutal-yamal-foto-video) located in Yamal-Nenzen Autonomous Okrug (YNAO) in the morning of 6th August 2021

Page 4 Line 11: Expanding on the previous comment, it is imperative that the authors conduct HYSPLIT back trajectories ending with an altitude that correspond to the sample collection height. While it is great to understand the evolution of the plume at an altitude of 500m AGL, it becomes challenging to link these observations with samples that were (presumably) collected at ground level. As such, additional trajectories need to be simulated to better represent the actual sample that was collected (e.g., 10m AGL), especially since many of the authors conclusions/interpretations rely on the description of long-range transport.

*Response: We thank the reviewer for addressing this issue. We studied the long-range transportation by the BWT calculations at 500 m AGL because this altitude is commonly used in the literature and regarded as the most representative. We agree that a study for the wider range including the ground level should be done additionally. In Supplementary Material (containing the figures below),*

the HYSPLIT calculations are presented for both Nadym (08/08/2021) and Bely Island (05/08/2021) at the range of altitudes of 10 m, 100 m, 250 m and 500 m. One can see that air mass transportation at any altitude including 10 m took place through the smoke plume covered the large territory of Western Siberia, BWT arrived to the station site from the same direction. Therefore, we show that calculations conducted for 500 m can be representative for the data presentation in this paper.

[Figure]

Page 6 Line 4: HYSPLIT typo

Response: The spelling is corrected to "HYSPLIT".

Page 6 Line 6: The use of language such as 'unprecedented' should not be done lightly. If such a qualifier is to be used, then the authors should provide some context for why this plume is indeed unprecedented. My overall recommendation would be to soften this though, as there are plenty of other examples of extreme smoke events worldwide (e.g., 2023 Canadian wildfires that blanketed significant portions of USA and Canada in smoke).

Response: We agree with the author that several extreme wildfires and smoke transportation, but we would like to emphasize the individual fire events connected to transportation into the Arctic ecosystem. As described by Tomshin & Solovyev (2022), summer 2021 had an extreme fire season in Yakutia, being the worst for four decades, especially from 24th July to 12th August 2021, caused by high air temperature and low precipitation along with a high-pressure system. (Tomshin and Solovyev, 2022) We were unable to find a comparable event that caused TC concentrations up to 150 μg/m$^{-3}$ 2000km distant from the BB source.

Ambient PM is continuously monitored at the Arctic Station on Bely Island, reporting weekly averages of organic aerosol <1μg/m$^{-3}$ and BC of approximately 50 ng/m$^{-3}$. Our study shows more than one order of magnitude higher concentrations of OC and EC for this intense fire season, which is also 10 times the typical variation between June and September of BC at Bely Island. (Popovicheva et al., 2023) In fact, for the region of the sampling site, this event was unprecedented.

Such efficient transportation of wildfire emissions to the Arctic is linked to a rather young phenomenon called circum-Arctic wave, which is an anticyclonic anomaly. (Yasunari et al., 2021) We expect that similar events will happen in the future again, but regarding our experience at Bely Island measurement station, it is justified to call it unprecedented.

Page 6 Line 14-17: Instead of using qualitative words such as 'almost' and phrases like 'clean artic air', I recommend that the authors improve the precision in this observation my using an actual metric, such as PM2.5, to support these statements more concretely.

Response: We agree with the reviewer's suggestion and modified page 6 line 14 to

*"When samples N07 and N08 were collected, OMPS aerosol index in Nadym had declined from >5 to < 0.625."*

and page 6 line 17 to

*"They brought clean Arctic air with an OMPS aerosol index below 0.625 from White, Barents and Kara Seas. The clean Arctic background for BC was previously determined from the 20th percentile of a 1.5 years continuous monitoring, which accounted for 10 ng m$^{-3}$. Background pollutant concentrations in Arctic stations are generally very low without any detectable influence from local or regional sources (Eleftheriadis et al., 2004; Popovicheva et al., 2019). Conversely, episodes of pollution were defined by the 80 percentile, accounting for 90 ng m$^{-3}$ (Popovicheva et al., 2022)."*

Page 6 line 30 – 35: this language is much more clear, but a somewhat repetitive version nonetheless of what was stated on Page 6 lines 14 – 17. I recommend merging this information for a more concise story.

Response: We intentionally created these two sections, in order to first outline the meteorological situation based on satellite data including BWT analysis and OMPS index, and secondly add results from chemical analysis.

Page 7 Line 18: The authors use sample identifies in the text (e.g., NO1), but dates in the figures. Please pick a consistent way to refer to sample to ensure easier comparison/interpretation between text and figures.

Response: We agree with the reviewer that there needs to be a consistent identifier for each sample. As in some cases, several samples were collected on the same day, only referring to a sample by its date is not sufficient.

Page 7 Line 20: Where is this data? If it is going to be presented and discussed, it needs to at least be shown in the SI.

Response: We thank the reviewer for this suggestion and added a figure to the SI. Furthermore, the typo of the correlation coefficient was changed from 0.69 to 0.59.

Page 8 Line 6 – 9: Now I have better context for the term 'unprecedented.' I would still recommend against its use overall though for reasons highlighted before. Again, an increase in concentrations by factors of ~10 is high, but likely not unprecedented given other worldwide wildfire aerosol observations.

Response: We agree with the reviewer that 10 times higher concentrations of particulate matter is not unprecedented compared to other wildfires in global context. However, as emphasized in a previous response, meteorological conditions were exceptionally favorable for generating such high concentrations in the Arctic, which was indeed unprecedented to the best of our knowledge.

Figure 3: A few of these mass spectra, particularly those for CHNO and CHO detected by APPI, CHNO detected by ESI- and CHO detected by ESI+ look as if they might be contaminated in the ~m/z 500 – 800 range. Are these major peaks that otherwise don't fit with the gaussian-like distribution separated by 44 Da? If so, there might be a significant PEG contamination issue. If yes, this is particularly concerning as PEG does not contain nitrogen – the authors peak assignments should be carefully inspected again. If these 'outliers' are instead fatty acids (high H/C, low O/C), the authors should consider manually removing them (while still describing it in the SI) as there is a strong likelihood that these are contaminant related, and not due to the sample. Further general info may be found here: https://beta-static.fishersci.ca/content/dam/fishersci/en_US/documents/programs/scientific/brochures-and-catalogs/posters/fisher-chemical-poster.pdf . This issue is perhaps further compounded/realized in Figure S5, most notbly in the top center and top right panels. Biomass burning samples are generally expected to exist as a bly in the top center and top right panels. Biomass burning samples are generally expected to exist as a continuum of species (e.g., https://pubs.acs.org/doi/full/10.1021/acsearthspacechem.1c00141), and therefore, the large 'gaps' in data here suggest that further data critiquing/cleaning is needed to ensure that all shown peaks are truly representative of the sample.

Response: We agree with the reviewer that some mass spectra from APPI, and ESI+/- show peaks that do not fit to the main Gaussian-like distribution. All mass spectra shown in the paper are blank corrected, with a field blank filter sample that was treated in the same way as the other filter samples. For blank correction, the peaks detected in the mass spectrum of the field blank were removed from the mass spectra of the samples. This is also the reason for the mentioned "gaps" in the DBE versus carbon number plots of some CHO compound classes in Figure S5, as these peaks were removed during blank correction due to their occurrence in the blank mass spectrum. We have added a more detailed description of the blank correction process to the SI.

We agree with the reviewer that especially in APPI compounds similar to typical contaminants in atmospheric pressure ionization are visible, however, as these compounds are not common in all samples and are not found in the blank spectrum, these compounds are a unique feature of the sample and it was decided against artificially removing single peaks from the data. Also, the discussion of the chemical composition is focused on relative numbers and not relative intensity of compounds, the error that may be induced by including possible contamination signals is expected to be negligible.

Figure 3-4: The authors show prominent entries for CHNOS peaks in Fig 4, but they are not shown in Figure 3. If the authors observed CHNOS, then a Figure 3-like representation of them should at least be added to the SI.

Response: We believe the reviewer is referring to Figures 4 and 5. As the number of CHNOS compounds is very low compared to the other displayed compound classes in Figure 4, except for ESI-, it is not well possible to display the CHNOS compound class together with the other classes in the Figure. We have added a mass spectrum of the CHNOS compounds in ESI- to the SI (Fig. S10).

Page 8 Line 35: while I understand that terms like 'lipid region' have been used historically to evaluate VK diagrams, it does not feel appropriate given the nature of the sample (i.e., it is not relevant to the sample

or research questions to refer to detected compounds as 'lipids' or even 'lipid like'). While I defer to the authors, my preference would be to avoid using such classifiers that aren't particularly relevant to the sample at hand.

Response: We agree with the reviewer that descriptive terms for areas in VK diagrams should be treated carefully, however the authors consider it helpful to use the terms, when implemented in broader context, to help readers that are not as familiar with VK diagrams. As in this paper the term lipid-region is accompanied by the detailed description of what kind of compounds are found in this area (H/C > 2, low O/C, low aromaticity, low carbon oxidation state) it should be sufficiently clear to the reader.

Page 8 Line 39: While true, this statement in and of itself is not novel. To be fair, the authors aren't saying it is https://pubs.acs.org/doi/full/10.1021/acsearthspacechem.1c00141, the same as referenced above) which have similarly shown the utility of a multi-modal assessment of wildfire particles.

Response: We agree with the reviewer and have added the mentioned reference as well as changed the sentence to more clearly state that this is part of the justification and motivation to use different ionization techniques.

Page 9 Line 9-10: The authors should clarify that by 'the highest abundant compound class,' they are referring to the class with the most assignments. Although it is likely that the CHO class also has the highest actual abundance, it cannot be definitively stated using uncalibrated MS data.

Response: We agree with the reviewer and have changed the sentence to be more precise.

Page 9 Line 16: Point the reader to a particular figure.

Response: We have added a reference to Figure 4b.

Page 9 Line 17-18: If a reference to 'the literature' is made, then it should be supported with appropriate references.

Response: We agree with the reviewer and have added specific references.

Page 10 Line 21: What is meant by 'most frequently detected'? Surely CHO compounds were detected in every sample too, so it isn't clear what this statement implies.

Response: The CHNO class showed the second highest relative intensity but the highest relative number of the compound classes. We have changed the sentence to be clearer.

Section 3.3.4.: If the potential contamination concerns outlined in a previous comment are true, then the authors should very carefully consider whether the peaks discussed in this section are truly endogenous to the sample, or perhaps an additional contamination artifact.

Response: We agree with the reviewer that possible contamination need to be considered during the discussion. However, as discussed in more detail in a previous comment, the blank correction with a

field blank was carried out for all samples. Also, compounds discussed in this section are most or uniquely found in the wildfire impacted samples. Therefore, there is a high certainty that these CH and CHN compounds originate from the wildfire and not from contamination.

Page 12 Lines 11 – 22: Direct the reader to Figure 6 at some point here.

We agree and have added a reference to Figure 6 to the text.

Page 13 Lines 14 – 16: While the correlations shown in Figure 6 appear to indeed show a trend, I caution against using this data in the current tone. Again, the authors do indeed seem to show a trend. But given that uncalibrated MS data is inherently not quantitative, that there is likely isomer effects, etc., multiple additional caveats needed to be stated before this data can be shown. The authors state in the SI that 'As REMPI is conducted under vacuum conditions, the intensity of an analyte is linearly proportional to its concentration in contrast to direct injection of samples and ionization under atmospheric pressure conditions, such as ESI or APPI.' This is not entirely true in its current wording, as MALDI and LDI (both vacuum based techniques) are well known to exhibit matrix effects. In summary: while the authors present the data interpretation in a way that may be appropriate to the study at hand, the implications should be clarified to better include caveats and need for calibration.

Response: We thank the reviewer for bringing up this topic. With the use of moderate laser fluences (in the order of $10^7$W m$^{-2}$), REMPI is in fact a quantitative technique (Boesl, 2000) and gives linear responses with increasing concentrations with a response factor known as photoionization cross section. (Gehm et al., 2018; Miersch et al., 2019) Like any other analytical technique, REMPI-TOFMS suffers from more than one analyte contributing to the same channel (i.e., *m/z*), which also cannot be solved by calibration. However, due to its high ionization selectivity, the number of possible interferences is even rather limited compared to other techniques.

The statement about linearity and comparison to AP techniques was intended to relate to the ionization techniques used in this study. When the pressure in the REMPI ion source is increased, it is shifted to medium- and atmospheric pressure laser ionization (MPLI/APLI). In this comparison, our statement remains correct as matrix effects and non-linear responses between signal and analyte concentration occur.

REMPI is a gas phase ionization technique, whereas LDI and MALDI are used for solid samples. Therefore, "vacuum condition" refers to a different meaning. During the ionization in MALDI and LDI, a plume from volatilized solid samples is generated with mean free path between molecules, atoms and ions shorter than in our REMPI ion source. Otherwise, no charge transfers could occur, which is the main mechanism for these techniques. Although MALDI/LDI can be carried out under vacuum conditions in terms of sample environment, the ionization does not take place under such. Considering the meaning of vacuum conditions, our statement remains generally true because due to the inverse relation between pressure and mean free path of particles during the ionization, no interactions between particles (ions, atoms and molecules) are possible.

Page 14 Lines 22 – 23: This statement (the link to proteins) needs to be referenced.

Response: We believe that the reviewer refers to page 15 Lines 22-23. We have added a reference for the statement. (Fuentes et al., 2010)

SI: The choice of a 1-1 methanol:dichloromethane mixture is interesting on the basis of comparison to other wildfire studies. The authors should comment in the manuscript that their chosen solvent conditions are more likely to bias the extraction to non-polar constituents. Can the authors also comment on their ESI spray stability after using such a high concentration of DCM?

Response: We agree with the reviewer that the chosen solvent mixture of MeOH/DCM is useful for the comparison to other wildfire studies. The authors consider this solvent mixture to be a broadband extraction that is able to extract polar as well as non-polar compounds. The selection of e.g., water or only methanol may slightly enhance the extraction efficiency of highly polar compounds, but as was shown in this study, highly polar compounds (e.g., CHO16, Figure S12) are also extracted by the chosen solvents.

The ESI spray stability was not negatively affected by the portion of 50% DCM. It was made sure that the ESI spray is stable before each measurement was started.

SI: were radicals allowed for in the APPI assignments?

Response: Yes, radicals were allowed for APPI assignments. We have added this to the text SI.

SI: Were blank mass spectra recorded? If so, how were they accounted for in data processing?

Response: Yes, field blank and solvent blank mass spectra were recorded and used for blank correction by deletion of signals in the blank spectrum from the spectra of the filter samples. As discussed for previous comments, an explanation of this was added to the text.

**References**

Boesl, U.: Laser mass spectrometry for environmental and industrial chemical trace analysis, J. Mass Spectrom., 35, 289–304, https://doi.org/10.1002/(sici)1096-9888(200003)35:3%3C289:aid-jms960%3E3.0.co;2-y, 2000.

Calì Quaglia, F., Meloni, D., Muscari, G., Di Iorio, T., Ciardini, V., Pace, G., Becagli, S., Di Bernardino, A., Cacciani, M., Hannigan, J. W., Ortega, I., and Di Sarra, A. G.: On the Radiative Impact of Biomass-Burning Aerosols in the Arctic: The August 2017 Case Study, Remote Sens., 14, 313, https://doi.org/10.3390/rs14020313, 2022.

Fuentes, M., Baigorri, R., González-Vila, F. J., González-Gaitano, G., and García-Mina, J. M.: Pyrolysis-gas chromatography/mass spectrometry identification of distinctive structures providing humic character to organic materials, J. Environ. Qual., 39, 1486–1497, https://doi.org/10.2134/jeq2009.0180, 2010.

Gehm, C., Streibel, T., Passig, J., and Zimmermann, R.: Determination of Relative Ionization Cross Sections for Resonance Enhanced Multiphoton Ionization of Polycyclic Aromatic Hydrocarbons, Appl. Sci., 8, 1617, https://doi.org/10.3390/app8091617, 2018.

Ikeda, K. and Tanimoto, H.: Exceedances of air quality standard level of PM 2.5 in Japan caused by Siberian wildfires, Environ. Res. Lett., 10, 105001, https://doi.org/10.1088/1748-9326/10/10/105001, 2015.

Johnson, M. S., Strawbridge, K., Knowland, K. E., Keller, C., and Travis, M.: Long-range transport of Siberian biomass burning emissions to North America during FIREX-AQ, Atmos. Environ., 252, 118241, https://doi.org/10.1016/j.atmosenv.2021.118241, 2021.

Lavoué, D., Liousse, C., Cachier, H., Stocks, B. J., and Goldammer, J. G.: Modeling of carbonaceous particles emitted by boreal and temperate wildfires at northern latitudes, J. Geophys. Res., 105, 26871–26890, https://doi.org/10.1029/2000JD900180, 2000.

May, N. W., Bernays, N., Farley, R., Zhang, Q., and Jaffe, D. A.: Intensive aerosol properties of boreal and regional biomass burning aerosol at Mt. Bachelor Observatory: larger and black carbon (BC)-dominant particles transported from Siberian wildfires, Atmos. Chem. Phys., 23, 2747–2764, https://doi.org/10.5194/acp-23-2747-2023, 2023.

Miersch, T., Czech, H., Stengel, B., Abbaszade, G., Orasche, J., Sklorz, M., Streibel, T., and Zimmermann, R.: Composition of carbonaceous fine particulate emissions of a flexible fuel DISI engine under high velocity and municipal conditions, Fuel, 236, 1465–1473, https://doi.org/10.1016/j.fuel.2018.09.136, 2019.

Narita, D., Gavrilyeva, T., and Isaev, A.: Impacts and management of forest fires in the Republic of Sakha, Russia: A local perspective for a global problem, Polar Sci., 27, 100573, https://doi.org/10.1016/j.polar.2020.100573, 2021.

Popovicheva, O. B., Chichaeva, M. A., Kobelev, V. O., and Kasimov, N. S.: Black Carbon Seasonal Trends and Regional Sources on Bely Island (Arctic), Atmos. Oceanic Opt., 36, 176–184, https://doi.org/10.1134/S1024856023030090, 2023.

Popovicheva, O. B., Evangeliou, N., Kobelev, V. O., Chichaeva, M. A., Eleftheriadis, K., Gregorič, A., and Kasimov, N. S.: Siberian Arctic black carbon: gas flaring and wildfire impact, Atmos. Chem. Phys., 22, 5983–6000, https://doi.org/10.5194/acp-22-5983-2022, 2022.

Semoutnikova, E. G., Gorchakov, G. I., Sitnov, S. A., Kopeikin, V. M., Karpov, A. V., Gorchakova, I. A., Ponomareva, T. Y., Isakov, A. A., Gushchin, R. A., Datsenko, O. I., Kurbatov, G. A., and Kuznetsov, G. A.: Siberian Smoke Haze over European Territory of Russia in July 2016: Atmospheric Pollution and Radiative Effects, Atmos. Ocean. Opt., 31, 171–180, https://doi.org/10.1134/S1024856018020124, 2018.

Tomshin, O.; Solovyev, V. Detection of burnt areas in Yakutia on long-term NOAA satellites data (1985-2015). Proc. SPIE 11560, 24th International Symposium on Atmospheric and Ocean Optics, 108338B, 2018

Tomshin, O. and Solovyev, V.: Features of the Extreme Fire Season of 2021 in Yakutia (Eastern Siberia) and Heavy Air Pollution Caused by Biomass Burning, Remote Sens., 14, 4980, https://doi.org/10.3390/rs14194980, 2022.

Yasunari, T. J., Nakamura, H., Kim, K.-M., Choi, N., Lee, M.-I., Tachibana, Y., and Da Silva, A. M.: Relationship between circum-Arctic atmospheric wave patterns and large-scale wildfires in boreal summer, Environ. Res. Lett., 16, 64009, https://doi.org/10.1088/1748-9326/abf7ef, 2021.

Yue, S., Zhu, J., Chen, S., Xie, Q., Li, W., Li, L., Ren, H., Su, S., Li, P., Ma, H., Fan, Y., Cheng, B., Wu, L., Deng, J., Hu, W., Ren, L., Wei, L., Zhao, W., Tian, Y., Pan, X., Sun, Y., Wang, Z., Wu, F., Liu, C.-Q., Su, H., Penner, J. E., Pöschl, U., Andreae, M. O., Cheng, Y., and Fu, P.: Brown carbon from biomass burning imposes strong circum-Arctic warming, One Earth, 5, 293–304, https://doi.org/10.1016/j.oneear.2022.02.006, 2022